# Catalytic Production and Upgrading of Furfural: A Platform Compound

**DOI:** 10.3390/ijms252211992

**Published:** 2024-11-08

**Authors:** Peng Gan, Kai Zhang, Guihua Yang, Jinze Li, Yu Zhao, Jiachuan Chen

**Affiliations:** 1Key Laboratory of Clean Pulp & Papermaking and Pollution Control of Guangxi, College of Light Industrial and Food Engineering, Guangxi University, Nanning 530004, China; ganpdyx@163.com; 2State Key Laboratory of Biobased Materials and Green Papermaking, Qilu University of Technology (Shandong Academy of Sciences), Jinan 250353, China; ygh@qlu.edu.cn (G.Y.); x1062310349@163.com (J.L.); yuzhao_77@163.com (Y.Z.)

**Keywords:** furfural, platform compound, C5 chemicals, reaction mechanism, actual reaction efficiency

## Abstract

Furfural is a renewable platform compound that can be derived from lignocellulosic biomass. The highly functionalized molecular structure of furfural enables us to prepare a variety of high value-added chemicals, which will help realize biomass high-value utilization, and alleviate energy and environmental problems. This paper reviews the research progress on furfural production and upgrading to C5 chemicals from the catalyst perspective. The emphasis is placed on summarizing and refining the catalytic mechanism and in-depth analysis of available data. Specifically, the reaction mechanism of furfural production and upgrading is summarized firstly from the perspective of reaction pathways and reaction kinetics. Then, the available data are further processed to evaluate the actual reaction efficiency of different catalytic systems from multiple dimensions. Finally, based on statistical analysis, the challenges and opportunities of furfural-based research are proposed.

## 1. Introduction

In the past century, the development and utilization of fossil resources have opened the door to the new world and made significant contributions to human economic prosperity and social progress. However, the rapid consumption of fossil resources has caused social problems such as resource shortages and greenhouse gas emissions [1]. With the emergence of fossil resource shortcomings, humans have to seek alternative energy to ensure normal production and life [2,3].

As a renewable organic resource, lignocellulose has given new impetus to the sustainable development of modern society. Lignocellulose fixes carbon dioxide in the environment to biomass through photosynthesis to provide human society with a large amount of sustainable biological raw materials and chemicals [4].

The lignocellulose biomass has a complex structure and stable properties, mainly composed of cellulose, hemicellulose, and lignin. For different plants, the percentages of these three parts vary greatly. Before their direct use as bio-feedstocks, pretreatment must be carried out with the aim of (1) breaking down the structure of lignocellulose; (2) decreasing the cellulose crystallinity; and (3) separating and preserving the complete structure of hemicellulose and lignin [5]. Once the lignocellulose is deconstructed, it can be converted to platform compounds, which can be further transformed into a variety of value-added chemicals [6].

According to the list of “Top 10 Biomass Derived Platform Compounds” revised by the US Department of Energy in 2010, furfural is considered to be a multifunctional bio-based C5 platform molecule due to its potential applications in biomass energy [7]. It is widely acknowledged that the C5 sugars derived from lignocellulose usually serves as raw material for furfural production, while the C6 sugars can be effectively converted into 5-hydroxymethylfurfural [8]. However, some studies found that the rapid pyrolysis of C6 sugar could result in the generation of furfural within the reaction system [9]. Although C6 sugars also possess potential for furfural production, the efficient catalytic conversion of C6 sugars into furfural remains a formidable challenge due to the inherent difficulties associated with C removal and isomerization [10]. The factors affecting the above reaction process mainly include the catalyst, reaction conditions and solvent system; meanwhile, the reaction conditions and solvent system usually need to be selected and optimized according to the catalytic activity of the catalyst [11]. Therefore, the catalyst is a vital point that affects the production of furfural from lignocellulose.

Many types of catalysts have been reported, which can be divided into homogeneous and heterogeneous catalysts according to the reaction system. Homogeneous catalysts are distributed uniformly in solvents and have high catalytic performance. They are in the same phase as the reactants, but separation and recovery have always been the key problems that restrict their development. Heterogeneous catalysts have an obvious phase interface with solvents, making them easy to recycle and reuse in the absence of solid biomass residues after the reaction, but they usually require external forces to strengthen the contact with the reactants. At present, great progress has been made in the research of catalysts in the furfural production process, but there are still some problems, such as harsh reaction conditions and poor catalyst performance. It is still necessary to explore new catalysts to solve them.

On the other hand, the molecular structure of furfural includes a furan ring with an aldehyde group, along with other functional groups, which can be converted into a variety of C5 chemicals through selective hydrogenation, hydrogenolysis, ring opening, and molecular rearrangement [12,13,14]. Now, although a higher selectivity of C5 chemicals has been obtained, there are still some problems in the furfural upgrading reaction. For example, the conversion mechanism is unclear. Therefore, it is still necessary to strengthen basic research to provide a theoretical basis for the catalytic upgrading of furfural. In addition, similar to furfural, biomass can be further dehydrated to produce 5-hydroxymethylfurfural after being hydrolyzed into hexose. 5-hydroxymethylfurfural is also a platform compound that connects biomass feedstocks with biomass refining. Compared to 5-hydroxymethylfurfural production, furfural offers significant advantages, including a broad range of raw materials, such as hemicellulose from agricultural and forestry residues, and relatively low production costs [15]. These characteristics give furfural greater potential in the chemical industry, making it a worthy focus for further research and development.

Although several important reviews articles on catalytic production and upgrading of furfural and/or 5-hydroxymethylfurfural have been published [16,17,18,19,20], a comprehensive overview of the production and upgrading routes of furfural, including the formation and causes of by-products routes, is currently lacking. The existing data should be comprehensively processed in order to accurately evaluate the actual reaction efficiency of the catalysts. In addition, the chemistries, processes and sustainability issues for the production and upgrading of furfural have not been systematically summarized in these works. Meanwhile, a lot of interesting results have been recently published on the trends and developments in catalytic production and upgrading of furfural. This manuscript mainly shows a comprehensive summary of the recent trends and developments in catalytic production and upgrading of furfural. The aspects of chemical reactions and industrial applications, such as the reaction conditions, solvent systems, the catalytic performance of existing catalysts, and possible mechanisms, are highlighted. More importantly, we conducted in-depth processing of existing data to comprehensively evaluate the actual reaction efficiency of the catalyst in terms of yield, selectivity, substrate loading, catalyst loading and reusability. In addition, the main achievements and the challenges still to be achieved in the pursuit of advancing the furfural-based research are proposed. Finally, the opportunities and suggestions for advancing furfural production and upgrading biorefineries are addressed in this review as well.

## 2. Main Feedstocks for Furfural Production

### 2.1. Conversion of C5 Sugars to Furfural

C5 sugars have the same number of C atoms as furfural and are the simplest feedstock for furfural production. Xylose, as a biomass-derived carbohydrate, can be simply isomerized and dehydrated to form furfural and obtain a high yield. Therefore, xylose is used as a substrate in the early research and development of new catalysts. Similarly, arabinose also shows excellent yield and selectivity in furfural production, which has been widely studied [21].

### 2.2. Conversion of C6 Sugars to Furfural

C6 sugars are a suitable feedstock for preparing 5-hydroxymethylfurfural [10]. However, recent studies have found that C6 sugars also have great potential in furfural production, such as glucose, fructose, etc. [9]. The difficulty lies in the removal and isomerization process of C elements. This is also a path for the development of the furfural industry in the future.

### 2.3. Conversion of Polysaccharides and Biomass to Furfural

The conversion of polysaccharides to furfural requires the process of hydrolysis. However, the hydrolysis of polysaccharides to monosaccharides is challenging because of the hydrogen bond restriction between molecules and the high number of by-products [22]. For example, the production of furfural from xylan requires Brønsted acid catalysis, with the generation of xylose, glucose, acetic acid, and so on. Similarly, directly converting biomass into furfural is more challenging. Generally, pre-treatment is required to obtain pre-treatment hydrolysate, such as corncob hydrolysate [23].

## 3. Catalytic Mechanism for Furfural Production

### 3.1. Reaction Pathway of Furfural Production

The furfural formation mechanism has always been a research topic for biomass high-value utilization. Currently, the conversion of biomass to furfural mainly consists of two steps: Brønsted acid catalyzes the hydrolysis of biomass to pentose, and pentose is further dehydrated to form furfural. The second step is more complex because it is a rate-limiting step [16].

The hydrolysis reaction refers to the process in which the β-1, 4-glycosidic bond is cut off and pentose is formed under the action of H^+^, which mainly includes four steps (Figure 1a) [16,24]: (1) The oxygen atom in hemicellulose is protonated. This process mainly exists in the oxygen atom of the glycosidic bond. However, due to the conformational restriction of hemicellulose along the glycosidic bond direction, the oxygen atom located in the pyranosyl ring may also be protonated. (2) After the oxygen atom is protonated, the sugar unit on the C1 position near the oxygen atom is isomerized to form a carbon cation, and the other sugar unit forms a hydroxyl group. (3) The water molecule reacts with the carbon cation to reconstruct the anomeric center and form hydroxyl groups. (4) Hydrolysis is continued until the disappearance of all oxygen bridges and the formation of monosaccharides. However, hemicellulose is a highly branched heteropolymer that can be hydrolyzed into pentose, hexose, and organic acids [25]. Pentose is the main substance that further dehydrates to form furfural, while hexose may be converted to 5-hydroxymethylfurfural through dehydration reaction. Acetic acid is the main by-product in the hydrolysis process, corresponding to the acetylation reaction of acetyl groups [25]. In addition, formic acid may also exist in the reaction system, as the aldehyde groups in furfural or 5-hydroxymethylfurfural may undergo cracking reactions.

After the hydrolysis reaction, the pentose will be further dehydrated and converted to furfural in the presence of Brønsted acid and nucleophile. In the homogeneous system, pentose is isomerized to 1,2-enediol intermediates and further dehydrated to form unsaturated aldehyde under the action of Lewis acid. Then, the unsaturated aldehyde undergoes dehydration and cyclization to form furfural under the action of Brønsted acid (Figure 1b) [26]. However, another dehydration mechanism exists when heterogeneous catalysts are used to produce furfural (Figure 1b). The process involves five steps [8,27]: (1) protonation of hydroxyl oxygen; (2) release of water molecules to form carbon cations; (3) breakage of the C-O bond to form C=O and C=C bonds, respectively, and release of a hydrogen ion; (4) repetition of steps 1 and 2, where hydrogen ions continuously attack the lone electron pair of the hydroxyl oxygen, producing two molecules of water, C=C bonds, and carbon cations; and (5) molecular cyclization through an elimination reaction, leading to the formation of furfural. Therefore, the catalyst plays a key role in the whole reaction network and even determines the formation mechanism of furfural. Similarly, there are side effects during dehydration. It mainly includes the condensation reaction of xylose and furfural and the degradation reaction of furfural.

In addition, hexose can also form furfural during the hydrothermal process. For example, furfural was observed during the pyrolysis of glucose [9]. There are two possible pathways for the conversion of hexose to furfural [28,29]. (1) Hexose dehydrates to form 5-hydroxymethylfurfural, followed by the loss of -CH_2_O to produce furfural. (2) Hexose is broken down into formaldehyde and pentose, which are then dehydrated to form furfural. Route 2 seems more reasonable. Zhang et al. did not detect furfural using 5-hydroxymethylfurfural as the raw material [9]. On the contrary, Cui et al. proposed that hexose could selectively break the C-C bond to form pentose [29]. 5-hydroxymethylfurfural is the main by-product in this reaction, as the activation energy for the preparation of 5-hydroxymethylfurfural from hexose is lower.

### 3.2. Reaction Kinetics of Furfural Production

The construction of kinetic models is essential for understanding and optimizing the production process of target products. In furfural production, the kinetic process primarily involves the acid hydrolysis and dehydration reactions of biomass resources, predominantly hemicellulose.

Firstly, during the acid hydrolysis stage, hemicellulose in biomass is hydrolyzed into monosaccharides such as xylose and arabinose under acidic conditions. The kinetic parameters of the hydrolysis reaction are primarily influenced by the reaction system. In homogeneous catalytic systems, hydrogen ions are uniformly dispersed, and the hydrolysis of hemicellulose is considered a first-order reaction, with the hydrolysis rate of oligosaccharides being faster than that of polysaccharides [30,31]. For instance, Kamiyama et al. noted that the hydrolysis rate of di-oligosaccharides was about 1.8 times that of penta-oligosaccharides [32]. In contrast, heterogeneous catalytic systems exhibit significant variations in the hydrolysis rate of hemicellulose. The hydrolysis rate is initially slow, accelerates during the middle stage, and then slows down again, eventually stabilizing. This behavior is more consistent with a second-order kinetic model, which is attributed to the changing collision probability between hemicellulose and active sites [33]. In heterogeneous systems, acidic sites are only present on the surface of the catalyst. The initial hydrolysis rate is determined by random collisions between hemicellulose molecules and acid sites [34]. As the reaction progresses, hemicellulose is hydrolyzed into oligosaccharides, increasing the collision probability with acid sites and consequently the hydrolysis rate. In the later stages of the reaction, the observed slowdown in the reaction rate can be explained by internal mass transfer limitations and steric hindrance, as smaller molecular substances may obstruct the pore structure of the heterogeneous catalyst [35]. 

In the dehydration stage, pentoses such as xylose and arabinose further dehydrate in an acidic environment to form furfural. The kinetic model for furfural formation is primarily related to side reactions involving furfural itself. Assuming that furfural does not participate in side reactions, the kinetic model for furfural formation aligns with that of hemicellulose hydrolysis, conforming to a pseudo-first-order model in homogeneous systems and a second-order model in heterogeneous systems [26,36]. However, in actual reaction processes, both condensation reactions involving xylose and furfural and degradation reactions of furfural take place. Therefore, a modified kinetic model is necessary to accurately represent the furfural production and consumption process.

Dehydration of hexose to produce furfural is more challenging owing to the higher energy requirement for C-C cleavage. The kinetic study showed that furfural production conformed to the pseudo-first-order kinetic model without considering side reactions [37]. However, side reactions are inevitable, and the kinetics of hexose dehydration to furfural still needs to be further studied.

## 4. Homogeneous Catalysts for the Production of Furfural from Biomass

A homogeneous catalyst refers to a catalyst that is in the same phase state as the reactant and acts independently in the form of a molecule or ion in the reaction system. Figure 2 illustrates the homogeneous catalysts commonly used in the furfural production process, including inorganic acids, organic acids, inorganic salts, and ionic liquids. Table 1 summarizes the research progress associated with these catalysts.

### 4.1. Mineral Acids

Mineral acids are widely used in catalytic reaction systems because of their excellent catalytic effect and low cost. 

Sulfuric acid is the first mineral acid used in the catalytic production of furfural. Quaker Oats used dilute sulfuric acid as a catalyst and obtained a furfural yield of approximately 50% at 153 °C with water steam heating for 5 h [56]. Most industries have reported similar furfural yields [57]. The main reason for the low furfural yield is that the hydrolysis rate of polysaccharides is much higher than the dehydration rate of monosaccharides, resulting in a large amount of monosaccharides accumulating and forming byproducts through homopolymerization and condensation reactions [58]. In order to further explore the reason for the low yield of furfural catalyzed by dilute sulfuric acid, Michocl et al. studied the mechanism of the hydrolysis of xylose to furfural catalyzed by dilute sulfuric acid on a time-resolved basis. They observed that at the beginning of the experiment, the content of open-chain xylose isomers was high (about 18% of total xylose), and then rapidly converted into products such as glyceraldehyde. Among them, xylopyranose could form the 2,5-acid anhydride of xylose and further dehydrate to form furfural [59]. Therefore, the catalytic production of furfural follows the Brønsted acid catalytic mechanism. In addition to sulfuric acid, researchers also tried to use nitric acid, hydrochloric acid, and phosphoric acid for the catalytic production of furfural. As strong mineral acids, nitric acid could promote the hydrolysis reaction. However, its promotion of the dehydration reaction was not enough, which could be due to the interaction between monosaccharides to form various by-products [60].

Compared to sulfuric acid, hydrochloric acid shows higher catalytic activity because chloride ion promotes the formation of 1,2-enediol [61]. As a weak acid, phosphoric acid exhibits excellent catalytic performance in dehydration reactions, but it is not suitable for hydrolysis reactions. On this basis, Oktay et al. systematically studied the effects of various mineral acids on the conversion efficiency of xylan. They found that mineral acids could promote furfural production, and the effects from high to low were in order as follows: hydrochloric acid, sulfuric acid, phosphoric acid, and nitric acid, which was related to the anionic composition of mineral acids [39].

### 4.2. Organic Acids

Organic acids are organic compounds with acidity. Most organic acids are lower in acidity than mineral acids, and have relatively low corrosive effects on equipment and cause less environmental pollution. Therefore, organic acids have been tested for use in furfural production. 

Formic acid is a common organic acid, which has been used as the catalyst in the production of furfural since 2009. Lamminpää et al. used formic acid as a catalyst to obtain a 60% furfural yield. The kinetic study showed that the decomposition of furfural was the main reason for the emergence of byproducts, and this phenomenon became more intense with increasing reaction temperature [62]. In the same year, Yang et al. confirmed that the catalytic activity of formic acid was higher than sulfuric acid and phosphoric acid in the production of furfural due to the adverse reaction caused by strong mineral acids [63].

As a kind of substance produced from the degradation of biomass, bio-based acids have the characteristics of environmental protection and nontoxicity. For example, Kim et al. studied the selective conversion of biomass hemicellulose into furfural by maleic acid under microwave heating conditions. Using pure xylose as the substrate, a 67% furfural yield could be obtained at 200 °C within 28 min. Using biomass xylose as the substrate, the furfural yield could reach up to 57% after the reaction at 200 °C for 28 min. Therefore, biomass xylose was more reactive to furfural formation than pure xylose. In addition, this team explored the reusability of maleic acid using biomass xylose as the research object. The results showed that after three consecutive cycles, the yields of furfural were 57%, 51%, and 48%, respectively, and the recovery rate of maleic acid could reach 85%. The reason for the decrease in furfural yield is the degradation of maleic acid to malic acid [42]. Methanesulfonic acid is considered a substitute for sulfuric acid because it has similar furfural selectivity to sulfuric acid. Rackmann et al. used methanesulfonic acid to catalyze xylose dehydration and achieved a 65% yield of furfural by reacting at 180 °C for 15 min. Interestingly, they found that the addition of glucose had a significant impact on furfural yield. It was possible that further reaction occurred with 5-hydroxymethylfurfural to form furfural [64].

As shown in Table 1, organic acids seem to exhibit higher actual reaction efficiency in furfural production compared to mineral acids. Perhaps the weak acidity of organic acids balances the hydrolysis and dehydration reactions of biomass, leading to a reduction in resinification. However, weak acidity also makes the dosage and catalytic conditions of organic acids slightly harsh.

### 4.3. Inorganic Salts

Inorganic salts have emerged as a research hotspot in the catalytic production of furfural because of their high catalytic performance and environmental friendliness. Chloride ion is widely employed in the production of furfural because it can promote the rotation process and the isomerization of biomass degradation [61]. However, the furfural yield did not meet expectations, which is closely related to the emergence of side reactions during the process of dehydration. Therefore, researchers tried to regulate the incidence of adverse effects by modifying the reaction system. Zhao et al. developed a green reaction system of CO_2_-water-isopropanol and used it to study the effect of NaCl on the conversion of xylose to furfural. The results showed that increasing the CO_2_ pressure would enhance the acidity of the reaction system, thereby promoting furfural production at an accelerated rate. Moreover, isopropanol could inhibit the formation of byproducts. In addition, the presence of NaCl would have no impact on the xylose conversion, while significantly enhancing furfural yield. The furfural yield was increased from 55.8% to 69.2% at 200 °C for 180 min [65].

In addition to chloride, sulfate is also used in the production of furfural. Al_2_(SO_4_)_3_ exhibited excellent catalytic performance in water-GVL biphasic solvents. The synergistic effect between Al_2_(SO_4_)_3_ and GVL was the key contributing to high catalytic activity. Al^3+^ was hydrolyzed into [Al(OH)_2_(aq)]^+^, which promoted the coordination/isomerization of xylose to xylulose. GVL could enhance xylan hydrolysis and xylose dehydration, expedite furfural formation and mitigate side reactions [49].

In addition, inorganic salts can also synergize with other acids to increase furfural yield. For example, at 145 °C, the furfural yield using HCl as a catalyst was 29%. The addition of CrCl_3_ increased the furfural yield to 39%, and the preparation time of furfural was also shortened by five times. By utilizing this catalytic system, a 76% yield (140 °C, 120 min) could be obtained in a two-phase system composed of water and toluene. Therefore, in the furfural production process, Lewis acids can catalyze the isomerization of xylose to xylulose, while Brønsted acids facilitate the dehydration of xylulose to furfural. The combination of the two would yield the greatest benefits [45].

### 4.4. Ionic Liquid

Ionic liquids, which are entirely composed of anions and cations in a molten salt form, have been used in furfural production due to their high stability, low corrosivity, and adjustable composition.

[bmim]SO_3_H and [bmim]HSO_4_ exhibit excellent catalytic performance in furfural production owing to their Brønsted acidity [66,67]. The anionic groups present in these ionic liquids not only act as Brønsted acids by providing protons, but also function as conjugated bases by accepting protons, thereby facilitating the alkylation of xylose and the subsequent dehydration to form furfural. It is noteworthy that the combination of ionic liquids and organic solvents can significantly enhance furfural yield because organic solvents can improve the catalytic activity of acidic protons and inhibit the occurrence of side reactions [66]. Xu et al. synthesized a series of [HSO_4_]-based protic Brønsted acidic ionic liquids and applied them to produce furfural in a two-phase system. They pointed out that the catalytic performance of ionic liquids was positively correlated with their acidity. Using [Hpy][HSO_4_] as the catalyst and water-MIBK as the solvent, 75.4% and 80.4% furfural yields could be obtained from xylose and hemicellulose by reacting at 180 °C for 120 min, respectively. More importantly, the catalytic system exhibited high reusability, and the furfural yield remained almost unchanged after 10 repetitions [68].

Lewis acidic ionic liquids are utilized in the production of furfural as well. Zhao et al. explored the effect of [bmim]Cl/AlCl_3_ on the conversion of arabinose to furfural in the butanone-water solvent system. In the reaction process, the addition of AlCl_3_ increased the selectivity of oxygenated aliphatics, while [bmim]Cl promoted the formation of cyclic ketones. In addition, they highlighted the significance of solvent effects on the production of furfural. The use of a butanone-water solvent would increase the selectivity of furfural [55]. Based on this premise, the team conducted molecular dynamics simulations to investigate the underlying mechanism behind butanone’s promotion of furfural production. The results confirmed that there was a competitive relationship between butanone and water molecules around xylose and furfural, which would promote the dehydration reaction of xylose and prevent the consumption of furfural. With a butanone to water volume ratio of 4:1 and [bmim]Cl/FeCl_3_ as the catalyst, a furfural yield of 60% was achieved [69].

Ionic liquids are potential catalyst and solvent in furfural production due to their high stability and reusability. However, the actual reaction efficiency of ionic liquids still needs to be improved. Because the amount of ionic liquids in the process of producing furfural is large and the reaction conditions are harsh (Table 1), it is still necessary to improve their application environment.

### 4.5. Metal Complexes

Metal complex catalysts are coordination compounds formed by transition metal ions or atoms bonded with organic ligands, imparting catalytic activity [70]. The key advantage of metal complex catalysts lies in their ability to finely tune catalyst properties by modifying the type and structure of metal ions and ligands, thereby achieving high yields and purity of the target product. However, their application in furfural production is relatively limited due to challenges in recyclability.

Ana et al. evaluated the catalytic performance of various metal complexes, specifically including H_3_PW_12_O_40_ (PW), H_4_SiW_12_O_40_ (SiW), and H_3_PMo_12_O_40_ (PMo), for the liquid-phase catalysis of D-xylose to furfural [71]. Using DMSO as the solvent and a reaction temperature of 140 °C, the tungsten-based metal complexes achieved furfural yields comparable to those of H_2_SO_4_ and p-toluenesulfonic acid catalysts after 4 h (58–67%), whereas PMo yielded less than half this amount of furfural. The study also highlighted the significant effect of solvent choice on catalytic performance: in DMSO, furfural yields ranked PW > SiW > PMo, while in an aqueous system, the order reversed to PMo > PW > SiW. Geonu et al. compared the catalytic performance of metal complexes, inorganic acids, and ion-exchange resins in furfural production, specifically evaluating PW, H_2_SO_4_, and Amberlyst 15 for alginic acid conversion [72]. The PW catalyst exhibited the highest catalytic activity, yielding a maximum of 33.8% furfural in a tetrahydrofuran/water co-solvent at 180 °C within 30 min. Based on product yield analysis over time, the authors proposed that furfural production involved the hydrolysis of alginic acid into monomers, followed by decarboxylation and dehydration to generate furfural. 

Given the limited catalytic activity of single-metal complexes in furfural preparation, researchers have explored bimetallic complexes to improve yields. Ana et al. synthesized Cs_x_H_3-x_PW_12_O_40_ (Cs_x_PW) based on PW and applied it to the liquid-phase catalysis of xylose to furfural [73]. The results indicated that the bimetallic complex Cs_x_PW exhibited higher catalytic activity than PW, likely due to a synergistic effect between W and Cs. Additionally, they supported Cs_x_PW on silica to obtain a solid acid catalyst, which demonstrated high selectivity for furfural production.

While issues like low yield and limited recyclability constrain the industrial use of metal complex catalysts in furfural production, advantages such as adjustable structure and tunable metal ions make them promising for furfural production.

## 5. Heterogeneous Catalysts for the Production of Furfural from Biomass

Heterogeneous catalysts have excellent recyclability, which depends on the absence of residual solid biomass or humin formation after the reaction. The primary heterogeneous catalysts employed in furfural production are solid acid catalysts, which typically possess ample pore structures and facilitate the optimization of substrate reaction conditions. Figure 3 depicts the commonly utilized types of solid acid catalysts, which encompass metal oxides, solid phosphates, zeolites, carbon-based catalysts, supported catalysts, and sulfonated polymers. Table 2 outlines the research progress associated with these catalysts.

### 5.1. Metal Oxides

Metal oxides are reported to act as promoters in furfural production, either by functioning as Lewis acids that facilitate the reaction or by serving as intermediates that stabilize the dehydrogenation of pentose.

Due to their modifiable structure, as transition metal elements, the oxides of Ti and Zr can serve as catalysts for promoting furfural production. The TiO_2_-ZrO_2_ composite catalyst exhibited superior catalytic activity attributed to its elevated specific surface area and acid-base property. In the catalytic process of furfural production, acidic sites could enhance hydrolysis and dehydration reactions, while alkaline sites would provide high reactivity for isomerization. It is worth mentioning that the key point affecting the catalytic activity of the TiO_2_-ZrO_2_ composite catalyst was the control of temperature during the preparation process. The crystalline phase of the composite catalyst is susceptible to change due to temperature, which in turn affects its acidity, alkalinity and reactivity [74]. On this basis, Zhang et al. prepared a highly acidic SO_4_^2−^/ZrO_2_-TiO_2_ catalyst with a porous structure and applied it to the production of 5-hydroxymethylfurfural and furfural. The results showed that the Lewis and Brønsted acids of the catalyst could undergo mutual conversion through H_2_O adsorption, resulting in synergistic catalysis. However, the yields of 5-hydroxymethylfurfural and furfural were not satisfactory, at only 30.9% and 54.3%, respectively, at 170 °C for 120 min. But the catalyst showed excellent reusability. It could be reused by simple filtration and calcination without affecting its catalytic activity [75].

In addition to TiO_2_ and ZrO_2_, other transition metal oxides (Nb_2_O_5_, CuO) are also often used in the catalytic production of furfural. Research showed that Nb_2_O_5_ exhibits superior activity and selectivity compared to typical Brønsted and Lewis acids in the preparation of furfural. The reasons comprised three aspects: (1) Nb_2_O_5_ exhibited mesoporous characteristics, facilitating the diffusion of xylose molecules towards acidic sites within the material. (2) Nb_2_O_5_ had the characteristics of Brønsted and Lewis acids at the same time, resulting in a strong synergistic effect. (3) The activation energy required for the conversion of xylose to furfural by Nb_2_O_5_ was lower. Based on this, Garcia et al. obtained a 50% furfural yield from D-xylose in a two-phase system using Nb_2_O_5_ as the catalyst [76].

Similarly, Zn-doped CuO nanoparticles (NPs) exhibit a highly defective structure, which provides an abundance of active sites. In addition, Zn and Cu could act as Lewis acid sites to promote the isomerization reaction in the production of furfural. Mishra et al. synthesized Zn-doped CuO nanoparticles (NPs) using the ultrasonic chemistry method. NPs could dehydrate xylose under mild conditions (150 °C, 12 h) and obtain an 86% furfural yield [77].

Due to the limited mass transfer, the actual reaction efficiency of metal oxides in furfural production is lower than that of homogeneous catalysts, which is reflected in the higher catalyst loading and the lower furfural yield [101]. Therefore, the development of metal oxide catalysts with porous structures is still the key.

### 5.2. Solid Phosphates

Solid phosphate is a promising candidate for catalyzing the production of furfural due to its simultaneous possession of Brønsted acidity from hydroxyl groups and Lewis acidity from metal ions. Currently, the solid phosphate catalysts commonly utilized encompass zirconium phosphate, niobium phosphate, and iron phosphate.

Mesoporous zirconium phosphate prepared by the hydrothermal method showed excellent catalytic performance in furfural production. The large specific surface area provided a large number of active sites, and the rich pore structure increased the diffusion flux of the substrate. The synergistic effect of Brønsted/Lewis acid sites facilitates ring opening, isomerization, and dehydration. Therefore, Cheng et al. used mesoporous zirconium phosphate as the catalyst to study its catalytic performance on the preparation of furfural from xylose in a two-phase system. The results showed that the catalytic performance of the catalyst was mediocre. The furfural yield was only 52% after a 2 h reaction at 170 °C. However, zirconium phosphate could be regenerated after heat treatment, and the activity was not affected [78].

Similarly, in niobium phosphate, the P-OH and Nb-OH groups acted as Brønsted acid sites while the Nb^5+^ ions acted as Lewis acid sites, synergistically promoting furfural production. In addition, the characterization of the reused niobium phosphate catalyst revealed that although the total number of acid sites on the used niobium phosphate was slightly weak and heterogeneous, both the Lewis and Brønsted sites remained exposed on the catalyst surface, which explains why this material exhibits extremely high reusability [79,102].

Iron phosphate has rich Lewis acid sites, including Fe^3+^ and hydroxylated species by hydrolysis [103], which endows it with remarkable catalytic potential in furfural production. However, iron phosphate lacks the Brønsted acid site, and thus researchers tried to improve the catalytic environment of iron phosphate to increase the yield of furfural. NaH_2_PO_4_ as a co-catalyst is a superior option, providing the necessary Brønsted acid for the reaction. However, the proportion of iron phosphate to NaH_2_PO_4_ should be strictly regulated, as an excess of Lewis or Brønsted acid sites may result in undesirable side effects. Xia et al. confirmed that when the ratio of FePO_4_ to NaH_2_PO_4_ was 10/1, the furfural yield could reach up to 92%. In addition, the catalytic system could also produce 5-hydroxymethylfurfural. The yield was 44% at 160 °C within 1 h [104].

In summary, when solid phosphate is used as the catalyst for furfural production, the furfural yield is still low, and the actual reaction efficiency needs to be improved. The addition of a co-catalyst can improve this situation, but the recycling of the co-catalyst will be a problem.

### 5.3. Zeolites

Zeolite is a solid acid catalyst with a well-defined pore structure, possessing strong acidity and chemical stability that endow zeolite with excellent catalytic performance and widespread application in furfural production.

The pore size of zeolite is a critical factor that impacts the yield of furfural. Macroporous zeolites facilitate substrate diffusion and expedite intermolecular material exchange. However, this process also results in accelerated furfural loss, reduces furfural yield, and even causes pore plugging and catalyst deactivation. Microporous zeolite exhibits an exclusion phenomenon that impedes reactant diffusion into the interior of the zeolite, thereby reducing its catalytic efficiency. Therefore, zeolite with a pore size close to the molecular size of xylose (6.8 Å) and furfural (5.5 Å) is the best catalyst [105].

The pore size of H-mordenite is about 5.8 Å–7 Å, which seems to make it an excellent catalyst to promote furfural production. Guerbuez et al. used H-mordenite as the catalyst to obtain 82% furfural (175 °C, 180 min) from xylose in a water-GVL solvent. Interestingly, this catalyst was capable of converting glucose into furfural, providing a novel approach for selectively transforming renewable biomass feedstocks [83].

Similarly, Wang et al. demonstrated that hexose could be dehydrated to form furfural. Using HZSM-5 zeolite as a catalyst, they obtained a 27.8% furfural yield from fructose at 150 °C for 1 h. More importantly, the results showed that the presence of Brønsted acid was beneficial to furfural production, while the pore structure played a crucial role in enhancing its selectivity. In addition, they noted that the furfural preparation process may result in side reactions such as condensation, etherification, nucleophilic attack, and electrophilic attack of furfural molecules [84].

Similarly, H-MCM-22 zeolite (5.5 Å–6.2 Å) and SAPO-18 zeolite (8.2 Å) were also employed for catalyzing furfural production, exhibiting satisfactory performance within a relatively short reaction time [82,85,106].

Zeolite can obtain higher furfural yields under mild conditions with less loading, and the actual reaction efficiency is relatively high. However, zeolite catalysts generally need to be combined with organic solvents. Therefore, the recycling and utilization of organic solvents will be a concern for researchers.

### 5.4. Carbon-Based Catalysts

Carbon-based materials are increasingly utilized in furfural production due to their rich pore structure, exceptional thermal stability, and cost-effectiveness.

Graphene has found extensive applications in catalysis owing to its flexible carbon interlayer connection, high chemical stability, and facile graft modification. Among them, sulfonated graphene oxide (SGO) has been demonstrated as an efficient catalyst for furfural production. The excellent catalytic performance of SGO can be attributed to the presence of a strong aryl sulfonic acid group, an oxygen functional group, and a large specific surface area [107]. However, the incorporation of other substances may reduce the catalytic activity of SGO. For example, Trung et al. synthesized a magnetic sulfonated graphene oxide (Fe_3_O_4_/SGO) and applied it to catalyze the production of furfural from bagasse. The furfural yield was only 14.07% (190 °C, 90 min), but, the catalyst exhibited facile magnetic separation and exceptional reusability properties [108].

Acid-functionalized activated carbon is utilized in catalyzing the production of furfural. Research has demonstrated that activated carbon treated with H_2_SO_4_ and HNO_3_ exhibits higher selectivity for furfural than H-mordenite, which can be attributed to the synergistic effects of the oxyacid group and the sulfonation group [109].

After sulfonation, biochar can efficiently catalyze the production of furfural. Deng et al. prepared biochar catalysts containing acidic groups through the carbonization and sulfonation of the corncob hydrolysate residue. The catalyst was employed for catalyzing the preparation of furfural from corncob pre-hydrolysate in a two-phase system, yielding 14% furfural at 170 °C with only 5 wt% xylose. In addition, the catalyst exhibited remarkable performance and excellent recoverability during regeneration [89]. In contrast, phosphotungstic acid-functionalized biochar (PA-FCB) has higher selectivity for furfural. Li et al. investigated the catalytic activity of PA-FCB in GVL/H_2_O using corncob as the raw material. The results showed that the furfural yield reached 96.06% at 468 K for 120 min. Py-FTIR and XPS showed that PA-FCB contained both Brønsted acid sites (-COOH) and Lewis acid sites (C=C, -OH), which could provide a high catalytic activity [110].

As well as zeolites, higher furfural yields can also be achieved using carbon-based catalysts with less loading (Table 2). More importantly, carbon-based catalysts have low production costs and strong recyclability, which possess great potential for application in furfural production.

### 5.5. Supported Catalysts

Supported catalysts are considered to be the best alternative for toxic liquid acid catalysts. Among them, supported catalysts prepared with silica as a supporter exhibit high stability, activity, and reusability in the furfural production process.

MCM-41 is a mesoporous silica material featuring a highly ordered pore structure (pore size 2–10 nm). After chemical modification, MCM-41 showed exceptional catalytic performance. The development of the solid acid catalyst using MCM-41 as a support can be traced back to 2005, when Dias et al. found that sulfonic acid-anchored MCM-41 was an effective catalyst for the conversion of xylose to furfural due to its rich mesoporous structure. These mesopores provided a pathway for the rapid diffusion of furfural, avoiding the degradation of furfural. However, the enrichment of byproducts in the catalyst led to the passivation of the active sites [111]. On this basis, this team loaded sulfated zirconia onto MCM-41 to synthesize a solid acid catalyst PSZ-MCM-41. The study noted that the catalytic activity of PSZ-MCM-41 was related to the sulfur content, and the decrease in catalytic performance during reuse experiments was attributed to sulfur loss [95].

The mesoporous SBA-15 possesses a two-dimensional hexagonal through-hole structure, which endows it with high selectivity towards furfural upon sulfonation. However, the catalyst has poor reusability as a result of the accumulation of byproducts. The catalytic activity can be restored by H_2_O_2_ treatment [96]. Similarly, Agirrezabal-Telleria et al. synthesized an SBA-15 catalyst supported by sulfonic acid groups with a tunable pore structure and used it for furfural production. They stressed that the pore structure played a crucial role in furfural yield. Catalysts with macropores facilitated xylose conversion by enhancing diffusivity, but this molecular mobility also led to yield loss reactions. Conversely, smaller pore sizes restricted xylose diffusion and resulted in lower furfural yields [112].

KIT-6 has abundant mesoporous properties and holds potential for furfural production. Thi et al. employed a sulfonic acid-functionalized mesoporous silica KIT-6 catalyst to achieve efficient and selective dehydration of xylose into furfural in the two-phase water/toluene system. Under the optimum reaction conditions (170 °C and 2 h), the conversion rate of xylose reached 97.5%, while the yield of furfural was 92.2%. The excellent catalytic performance of the catalyst can be attributed to its three-dimensional structure, which facilitates the diffusion of xylose and furfural. Furthermore, the high acid density also impacts catalytic activity by potentially impeding reactant access to active sites [97].

### 5.6. Sulfonated Polymers

Sulfonated polymers are typical Brønsted acid catalysts, which have been used in the preparation of various biofuels. They are suitable alternatives to other solid acid catalysts due to their low cost and recyclability.

Sporopollenin and polytriphenylamine have stable structures and are easy to modify. After sulfonation, they can obtain excellent catalytic activity in furfural production. Wang et al. used sulfonated sporopollenin (SSP) as a catalyst in a water-CPME system to obtain 69% and 37% furfural yields from xylose and xylan, respectively. More importantly, the catalyst had excellent reusability and could be cycled ten times without losing its catalytic performance [99]. Similarly, Zhang et al. prepared a solid acid catalyst (SPTPA) using polytriphenylamine as a supporter and sulfonated it with chlorosulfonic acid. At 175 °C, 74% furfural and 32.3% 5-hydroxymethylfurfural were obtained from corncob using SPTPA as the catalyst [113].

In addition to modification, the direct preparation of sulfonated polymers using sulfonic compounds as organic units is also highly favored by researchers in furfural production. Xu et al. developed a solid resin acid (PTSA-POM) by copolymerizing paraformaldehyde and p-toluenesulfonic acid, and its waterproofing was significantly improved by high-temperature calcination. The use of PTSA-POM as a catalyst for furfural production in aqueous solution was first proposed. Under the optimal conditions, the furfural yields of corn straw and xylose were 80.4% and 83.5%, respectively. Compared with mineral acids, the catalytic efficiency of PTSA-POM was higher because there are no free hydrogen ions in PTSA-POM, which inhibit the degradation reaction of furfural. However, the reusability of the catalyst needed to be improved. After multiple reuses, the furfural yield and the xylose degradation rate both slightly decreased, which was closely related to the SO_3_H groups leaching [98]. In addition, the preparation process of covalent organic frameworks (COFs) also includes a large number of sulfonic acid-based organic monomers, which seem to have great potential in biomass conversion. It was reported that a high yield of 5-hydroxymethylfurfural could be obtained by using SO_3_H-COF as the catalyst [114]. However, the application of SO_3_H-COF catalysts in furfural production still needs to be studied.

## 6. Energy Challenges and Energy-Saving Strategies in the Furfural Industry

At present, the furfural industry still relies on relatively outdated and inefficient processes [115]. These have hindered the development of furfural and undermined its competitiveness in the alternative fuel market. In this section, we provide a brief overview of the furfural production process, emphasizing energy-intensive steps, potential areas for energy optimization, and strategies to reduce preparation costs.

### 6.1. Energy-Intensive Steps Within the Production Process

In 1921, Quaker Oats founded the world’s first enterprise to produce furfural [56]. They proposed using agricultural waste such as corn cob, oat husks, and sugarcane bagasse as raw materials for furfural production. After a century of development, modern industrial production of furfural still employs high-temperature and high-pressure reaction conditions [116]. Specifically, the process involves pre-treatment of biomass, including impurity removal and grinding. This is followed by mixing the pretreated material with mineral acids to generate furfural in a high-temperature and high-pressure environment. Subsequently, the furfural-laden steam discharged from the reactor is conveyed to a fractionating column, where it undergoes initial distillation and azeotropic evaporation, facilitating the separation of furfural, water, and light components. Finally, the addition of sodium carbonate solution eliminates acidic substances, resulting in a furfural solution with a purity exceeding 98.5%.

The substantial energy consumption in the furfural production workshop is a consequence of the lifting process required to overcome the structural limitations of biomass raw materials. This process involves sequentially achieving hydrolysis, ring opening, dehydration, cyclization, and other steps [107]. Secondly, the purification workshop for furfural also requires high energy to achieve the separation of furfural from other components.

### 6.2. Potential Areas for Optimizing the Furfural Industry

Facing the challenges of furfural industrialization, we conducted an in-depth analysis of the potential optimization fields of the furfural industry from four aspects: (1) insufficient furfural yield in the production workshop, the hydrolysis of biomass to pentose and the dehydration of pentose to furfural occur concurrently, leading to unfavorable condensation reactions; (2) excessive water consumption the pretreatment of biomass, the steam transportation of furfural, the condensation and separation of components all require a large amount of water as the medium, hindering the advancement of green production of furfural; (3) substantial waste of biomass components although pentosan constitutes the primary component of biomass used for furfural production, cellulose and lignin are merely utilized for combustion as furfural residue, resulting in a significant waste of resources [117]; (4) inadequate utilization of carbon dioxide during the alkali neutralization stage, sodium carbonate reacts with acidic substances to generate carbon dioxide, a primary contributor to the greenhouse effect. Considering these factors, the energy optimization of furfural industry should focus on the structural adjustment of production and separation workshops.

### 6.3. Strategies for Reducing Preparation Costs

In view of the challenges faced by the furfural industry, we propose several recommendations for achieving sustainable development in this field. (1) The implementation of a “two-step” furfural production strategy can enhance yield by conducting biomass hydrolysis and pentose dehydration separately. This approach not only minimizes the condensation of pentose and furfural but also reduces energy consumption. (2) Employing advanced technology can help to reduce energy consumption. An organic system could be used instead of a steam system for furfural separation, and membrane separation technology could be employed for furfural solution purification [118]. (3) Establishing a comprehensive biomass utilization industry chain will enable the high-value utilization of all biomass components during the furfural production process.

## 7. Catalytic Upgrading of Furfural to C5 Chemicals

As a platform compound, furfural can be selectively hydrogenated to prepare a variety of C5 chemicals, including furfuryl alcohol (FA), tetrahydrofurfuryl alcohol (THFL), pentanediol (PDO), 2-methylfuran (MF), and γ-valerolactone (GVL) (Table 3).

As shown in Figure 4, the upgrading of furfural involves at least one step, which includes hydrogenation (hydrogenation of double bonds), hydrogenolysis (breaking of single bonds), ring opening reaction, or molecular rearrangement. The selection of catalyst and support in the above process can affect the selectivity of the target product and even alter the reaction mechanism of furfural upgrading.

Metals and metal oxides are frequently employed as catalysts in the upgrading of furfural, effectively preventing excessive hydrogenation of furan rings, and enhancing the yield of the target product. In addition, metallic elements possess distinct geometric and electronic properties that can alter the catalytic mechanism of furfural upgrading [167,168].

The selectivity of the target product is primarily influenced by the support due to steric hindrance between the substrate and the pores. In addition, the surface properties of the support also impact catalyst stability through the metal-support adsorption mode, intermediate formation, and H spillover rate [169].

Based on this, this section will provide a summary of the research progress of furfural upgrading to C5 chemicals from the perspective of metal catalyst and support selection.

### 7.1. Furfuryl Alcohol (FA)

FA is considered to be the most important derivative of furfural, accounting for approximately 65% of its consumption [170]. According to different reduction methods, the catalytic hydrogenation of furfural to FA can be classified into two categories: catalytic hydrogenation and catalytic transfer hydrogenation (CTH).

Taking CTH as an example, Figure 5 shows the catalytic upgrading mechanism of furfural to FA using a metal oxide catalyst [128,171]. It mainly includes four processes: (1) the metal oxide acts as Lewis acid to adsorb O and H from the hydroxyl group of the hydrogen donor, thereby converting it into the corresponding alkoxide while simultaneously releasing the proton H; (2) metal ions adsorb O in the furfural aldehyde group to activate it; (3) the α-H and proton H in the hydrogen donor are adsorbed on C and activated O in C=O of furfural, respectively, forming a six-membered ring intermediate; (4) FA and ketones are formed from the intermediate through intra-ring hydrogen transfer.

Catalytic hydrogenation is a conventional approach for the synthesis of FA from furfural, which involves gas-phase and liquid-phase hydrogenation. Cu-based catalysts are commonly employed to produce FA in gas-phase hydrogenation due to the ability of Cu species to interact with defect sites in metal oxides, thereby enhancing their activity. The activation of O atoms in C=O bonds is achieved through the utilization of lone pair electrons, leading to the formation of alkyl intermediates [119,172]. In addition to gas-phase hydrogenation, Cu-based catalysts are often used for liquid-phase hydrogenation. Katarína et al. investigated the impact of Cu on the selectivity of FA in Pd-Cu catalysts and emphasized that metal loading and support selection were crucial factors in FA preparation. Different Cu loading can impact both the single metal Pd^0^ site and the bimetallic Pd^0^-Cu_2_O catalytic site, ultimately affecting carbonyl polarization and hindering hydrogen transfer from the adjacent Pd-H site. Among all bimetallic Pd-Cu catalysts, those supported on MgO or Mg(OH)_2_ exhibit superior catalytic performance. After reaction at 110 °C for 80 min, furfural was completely converted and the FA selectivity was higher than 98% [122]. 

In addition to Cu, Fe is also used for liquid-phase hydrogenation of furfural. Li et al. prepared a variety of Fe-doped Ni-B amorphous catalysts (Ni-Fe-B) and applied them to prepare FA from furfural by liquid-phase hydrogenation. The results showed that the addition of Fe significantly increased the catalytic activity. The FA yield increased from 55% to 100% (100 °C, 4 h) when the Fe content was 51%. The reasons can be summarized as follows: (1) as a dopant, the addition of Fe resulted in a more homogeneous distribution of the Ni active sites; (2) the Fe^3+^ ion activated the oxygen in the carbonyl group and enhanced the adsorption of the C=O bond onto the catalyst; (3) an electron transfer phenomenon occurred between Fe and Ni, which enriched Ni electrons and activated the C=O bond for the hydrogenation reaction [121].

CTH is a process wherein certain organic compounds act as hydrogen donors in the presence of catalysts, releasing hydrogen quantitatively and facilitating the hydrogenation reaction. The hydrogen donor involves formic acid, isopropanol, and other alcohols, and the catalyst mainly involves Cu, Pt, Fe, etc.

Du et al. employed CuO-Pd/C as a catalyst with formic acid as a hydrogen donor to achieve a 98.1% FA yield at 170 °C for 3 h. They pointed out that in this catalytic system, Pd facilitated hydrogen adsorption and promoted furfural conversion, nano-Cu enhanced FA selectivity, and the Cu-Pd alloy exhibited synergistic effects [125]. Similarly, there is a synergistic effect between Pt and Mg elements in the preparation process of FA. Zhang et al. prepared a novel catalyst Pt@MT-450 by loading Pt nanoparticles onto MgTiO_3_, and achieving efficient in situ hydrogenation of furfural to FA. During the reaction process, Pt and Mg jointly promoted methanol reforming to release H_2_. H_2_ partially participated in the hydrogenation reaction of furfural. The other part reduced Pt to Pt^0^ to improve the reusability of the catalyst. The catalytic results showed that the FA yield could reach up to 93% at 120 °C for 1 h. In addition, Pt@MT-450 exhibited stable catalytic performance and could be reused multiple times [173].

The pore size in the catalyst is also a key factor affecting the FA selectivity. Large-scale channels are conducive to molecular diffusion and accelerate the reaction, but excessive hydrogenation may occur and reduce the FA selectivity. MOFs are considered as ideal supporters with molecular size screening function, and have great potential in FA production [174]. Long et al. synthesized the catalyst Pt-CeO_2_@UIO-66-NH_2_ and obtained a high yield of FA. The excellent catalytic performance of the catalyst was attributed to the improvement of the catalytic activity of CeO_2_, and the channel size of UIO-66-NH_2_ limited the excessive hydrogenation of furfural [175]. Similarly, Jiang et al. prepared a MOF-derived magnetic Fe_3_O_4_/C catalyst, which obtained a 75% FA yield at 200 °C for 4 h, and exhibited excellent reusability. They emphasized that the excellent catalytic performance of the catalyst was inseparable from Fe_3_O_4_. In this process, Fe_3_O_4_ acted as Lewis acid, adsorbing O and H atoms in the hydroxyl group of isopropanol, and forming the corresponding alkoxide and proton H. The Fe^3+^ ion activated the O atom in C=O and promoted hydrogen transfer to form FA. Furthermore, they also pointed out that the reduction potential of the hydrogen donor and the space effect of the alkyl chain would affect FA yield [128].

### 7.2. Tetrahydrofurfuryl Alcohol (THFL)

The two-step process utilizing FA as an intermediate is capable of producing THFL, a green solvent widely used in the industrial field, from furfural. 

At present, there is not yet scientific consensus for the mechanism of preparing THFL through a one-step furfural process. The main disagreement is over the location of hydrogenation. Some researchers believe that hydrogenation first occurs at the C=O position of furfural, which generates FA [130]. Then, the furan ring is adsorbed parallel to the metal surface, causing C=C to undergo a hydrogenation reaction, forming THFL (Figure 6a). However, other researchers hold the opposite view, believing that the hydrogenation reaction first occurs in the furan ring structure (Figure 6b).

Ni-based catalysts exhibit excellent catalytic performance in the hydrogenation of furfural to THFL. For example, the Ni-Cu catalyst had synergistic effects in the catalytic process. The presence of Ni was conducive to the adsorption and activation of molecular hydrogen, leading to the formation of Cu(0)/Cu species in situ, which served as a prerequisite for activating the C=O bond in furfural and producing FA. In addition, the involvement of Ni species was beneficial for the further conversion of FA to THFL [130]. On this basis, Wu et al. also reached the same conclusion. A variety of Cu_x_Ni_y_/MgAlO catalysts were synthesized and applied to the hydrogenation of furfural. The results showed that the single-metal Cu catalyst could only selectively hydrogenate the aldehyde group to produce FA (100% yield), while both the bimetallic Cu-Ni catalyst and monometallic Ni catalyst could realize continuous two-step hydrogenation to form THFL. Finally, when the Cu/Ni ratio was 1/1, and the reaction condition was 150 °C for 3 h, the THFL yield reached 95%. The CuNi/MgAlO catalyst could be reused up to six times without affecting catalytic activity [133].

Based on the metal Ni, Su et al. proposed a novel method for preparing THFL by loading the metal Ni onto an MOF material, thermally decomposing it to gradually pyrolyze and shrink the organic ligand, and finally forming a Ni/C nano-catalyst. The interaction between pyrolytic carbon and Ni ions in the catalyst could provide active sites and uniformly dispersed Ni nanoparticles [131]. On this basis, Wang et al. synthesized a Ni@C@CNT catalyst through the pyrolysis of Ni-MOF doped with carbon nanotubes. They pointed out that Ni_3_C exhibited orbital hybridization phenomenon, resulting in better catalytic performance compared to transition metal catalysts in furfural hydrogenation, hydrodeoxygenation, and isomerization. Furthermore, Ni_3_C could adjust the adsorption strength of furfural on the active site, improving the selectivity of THFL. In terms of mechanism research, both Ni metal and Ni_3_C could activate the double bonds (C=O, C=C) of furfural to facilitate the conversion of furfural to THFL. Moreover, pore structure plays a crucial role in catalytic processes as a high surface area and a mesoporous structure enable more active sites for reaction [135].

Pd-based catalysts are also often used for the synthesis of THFL. Liu et al. developed a Pd-Ni/MWNT catalyst for the selective hydrogenation of furfural to THFL, where Ni facilitated the exposure of more active crystal planes of Pt (111) during the catalytic process, thereby enhancing the adsorption and activation of the C=O and C=C bonds. Moreover, this catalyst exhibited high stability and retained its superior catalytic activity after five cycles [176].

The catalytic activity of the bimetallic Pd-Ru catalyst in the preparation of THFL is comparable to that of the bimetallic Cu-Ni catalyst. The metal Ru was similar to Cu in that it could only adsorb and activate the C=O bond. Pd could activate the C=O bond and the C=C bond to form THFL. When used together, the yield of THFL would be higher. In addition, kinetic studies showed that tetrahydrofurfural serves as an intermediate in the catalytic system, where hydrogenation initially occurs at the C=C bond of the furan ring. The saturation of the furan ring significantly enhances the hydrogenation of the C=O bond [177].

### 7.3. Pentanediol (PDO)

PDO is a widely used chemical in the production of plasticizers, cosmetics, fungicides, and other products [178]. Although PDO can be obtained through the hydrogenation and ring opening of furfural, its synthesis technology has not yet reached maturity. Therefore, efficient preparation of PDO still has important research value.

FA is still used as an intermediate in the conversion of furfural to either 1,2-PDO or 1,5-PDO via selective opening of the FA ring. The formation mechanism is shown in Figure 7. Taking 1,2-pentanediol as an example (Figure 7a), the alkaline sites of the catalyst adsorb the -OH in FA, and the active hydrogen species at the metal site carries out a semi-hydrogenation reaction at the C4=C5 bond in the furan ring, in order to establish p,π-conjugation between the orbital of O1 and the C2=C3 bond, thereby weakening and breaking the C5-O1 bond, and generating an intermediate 1-hydroxy-2-pentanone. Then, the intermediate is subsequently subjected to C=O hydrogenation, leading to the formation of 1,2-PDO. The catalytic mechanism for 1,5-PDO is similar to that of 1,2-PDO. The C2=C3 bond undergoes a semi-hydrogenation reaction with active hydrogen species, resulting in the cleavage of the C2-O1 bond and forming 1,5-PDO after the hydrogenation reaction (Figure 7b).

Metal elements serve as common catalysts in the process of upgrading furfural to 1,2-PDO. For example, Pt nanoparticles supported by hydrotalcite exhibited a 73% yield during the 1,2-PDO preparation process. In this catalytic system, furfural was hydrogenated to FA by the Pt catalyst. Subsequently, the basic sites on the surface of hydrotalcite adsorbed and fixed FA. Finally, the active hydrogen species attacked the adsorbed furan ring, broke the C5-O1 bond and generated 1,2-PDO through a hydrogenation reaction [138]. Similarly, the bimetallic Ru_3_Sn_7_ nanoalloy on the ZnO catalyst (Ru-Sn/ZnO) also showed a high yield of 1,2-PDO. The basic ZnO support played a crucial role in promoting the formation of the active Ru-Sn alloy phase. Structural analysis revealed that SnO_2_ existed in the catalyst, which would facilitate the hydrolysis of furfural to produce FA. The basic support of ZnO could strongly interact with the hydroxymethyl moiety of FA, resulting in the formation of an alkoxide that facilitates hydrogenolysis to produce 1,2-PDO [143].

In 2011, Xu et al. first reported the successful application of a metal-based catalyst (Pt/Co_2_AlO_4_) for the direct upgrading of furfural to 1,5-PDO under mild conditions. The catalytic process was found to be synergistic between CoO_x_ and Pt, CoO_x_ was responsible for the absorption and ring-opening of C=C bonds, while Pt played a crucial role in the subsequent hydrogenation. Moreover, this study suggested that elevated temperatures could induce pentadiol isomerization to produce 1,2-PDO or 1,4-PDO [136].

Similarly, the abundant pore structure of the support facilitates the preparation of PDO. Yeh et al. embedded an alumina-supported platinum catalyst in an MOF to prepare the MIL-53-NH_2_ (Pt, Al) catalyst for 1,5-PDO synthesis. The results indicated that the Pt nanoparticles were highly loaded and uniformly distributed on the catalyst, which possessed abundant Lewis and Brønsted acid sites, and the Brønsted acid sites were derived from the penta-coordination aluminum. The structure and properties of the catalyst determined the excellent catalytic performance in the preparation of 1,5-PDO. In this catalytic system, sodium borohydride facilitated hydrogen proton separation and furfural hydrogenation to FA under the synergistic Lewis and Brønsted acid action, which was further converted to 1,5-PDO. In this process, NaBO_2_, the hydrolyzed product of NaBH_4_, was identified as a crucial factor in facilitating the hydrolysis of furfural to 1,5-PDO at temperatures close to ambient conditions [139].

### 7.4. 2-Methylfuran (MF)

MF is regarded as a substitute to gasoline because of its superior octane rating and energy density [179]. At present, the production of MF by the furfural hydrogenation-deoxidation process has generated great interest.

The use of FA as an intermediate in furfural upgrading to MF is a common practice. Therefore, this section will focus on the preparation mechanism of MF using FA as the starting material. As shown in Figure 8, there are two primary pathways for the conversion of FA to MF. The first pathway is the direct route (Figure 8a). The metal sites serve as adsorption sites for hydrogen protons, which then initiate an attack on the C1 site, leading to the cleavage of the C-OH bond and formation of MF [148]. The dissociated -OH moiety subsequently combines with another hydrogen proton to form water molecules. The second pathway involves activation of the aromatic rings (Figure 8b). It mainly involves two steps: (1) proton hydrogen adsorbed on the metal sites attack the C3 site, leading to transfer of the double bond between C2–C3 and C1–C2 and the cleavage of the C-OH bond to form unstable intermediates. (2) Another proton hydrogen attacks the C1 site, restoring the double bond between C2–C3 while extracting a H atom from C3 to form MF. The dissociation of H in C3 can occur through either proton hydrogen dissociation or original hydrogen dissociation, as both types are equivalent. It is noteworthy that ring activation serves as the primary means for converting FA into MF, with the separated H combining with the cleaved -OH to produce water molecules.

The process of upgrading furfural to MF involves both hydrogenation and hydrolysis. Similar to the FA production process, the catalytic methods include catalytic hydrogenation and CTH. The Cu-based catalyst is also suitable for upgrading furfural to MF, with Cu/SiO_2_ exhibiting high catalytic activity due to the strong interaction between dispersed Cu ions and SiO_2_. Driven by this interaction, the Cu^+^ sites and the weak acid content on the catalyst surface would increase, resulting in a synergistic effect that promotes the activation of the CHO group in furfural and the cleavage of the C-OH group in furfural. In addition, this study indicated that Cu particles with larger sizes dispersed on the support exhibited the highest selectivity to FA, resulting in a reduction in MF production. The Cu/SiO_2_ catalyst prepared by anchoring Cu nanoparticles onto a layered silicate structure showed a high selectivity for MF. In the catalytic process, silicates enhanced the Cu dispersion, thereby promoting the hydrogenation and hydrogenolysis of furfural. It is worth mentioning that the regulation of the Cu-Si molar ratio in the catalyst was very important. Excessive use of Cu led to its aggregation into CuO particles that covered the surface of the layered silicate, while the excess silicon resulted in a shortage of Cu acid sites, which weakened the selectivity of MF [148].

In addition to Cu, Pt-based catalysts are also used for MF preparation. Luo et al. employed Pt/C as a catalyst to investigate the effect of H_2_ pressure on the selective conversion of furfural to MF. The results showed that the MF was highly sensitive to H_2_ pressure. At low pressure, decarbonization was the main reaction pathway and the yield of MF was low, but furan selectivity could reach over 70%. Under high pressure, the hydrodeoxygenation of furfural proceeded through a sequential reaction network with MF as an intermediate [180].

The Cu-based catalyst also showed remarkable catalytic activity in the process of CTH of furfural for MF preparation. Niu et al. found that Cu-Zn and Zn-Al bimetallic catalysts demonstrated excellent catalytic performance by activating the C=O bond in furfural and promoting isopropanol dehydrogenation, thereby enhancing FA selectivity. On this basis, the utilization of a Cu-Zn-Al trimetallic catalyst was proposed for furfural catalysis, exhibiting significant furfural conversion (99%) and MF selectivity (72%). The Al component in this catalyst facilitated the dispersion of Cu species on the surface, while the Zn component provided surface reduction of Cu^0^ and Cu^+^ species, promoting the formation of CuAl_2_O_4_, which increased Lewis acid sites in the catalyst and improved the yield of MF [156].

In addition, researchers are increasingly focusing on the impact of pore size on MF yield. According to the steric hindrance effect, an appropriate pore size can improve the selectivity to target products and reduce the formation of byproducts. Seyeon et al. synthesized a mesoporous Cu-Al_2_O_3_ catalyst with exceptional hydrodeoxidation performance for furfural. They emphasized that the distinctive pore structure would improve the accessibility and dispersion of Cu and provide more Cu^0^ and Cu^+^ species. This would improve the weak acidity of the Cu-Al_2_O_3_ catalyst and boost its catalytic performance [181].

### 7.5. γ-Valerolactone (GVL)

GVL possesses excellent physical and chemical properties, making it highly applicable in organic synthesis, bio-refining, and petrochemical industries. Traditionally, the conversion of furfural to GVL requires levulinic acid as an intermediate; however, this process is time-consuming, labor intensive, and contradicts sustainable development strategies. Therefore, the direct upgrade of furfural to GVL is widely favored by researchers.

Currently, the mechanism of upgrading furfural to GVL is still unclear, and the widely accepted mechanism is shown in Figure 9 [182]. FA serves as the intermediate product in the conversion process from furfural to GVL. Therefore, FA is used as the initial substance to elucidate the mechanism of GVL production in this section. As shown in Figure 9a, under the combined action of Brønsted and Lewis acid, alcohols (hydrogen donor) react with FA to eliminate water and form intermediate I. Then, the intermediate Ⅰ bound water molecule undergoes isomerization into intermediate Ⅱ by the combined action of Brønsted and Lewis acids in an alcohol environment. Finally, proton H attacks the C=O bond and eliminates alcohol to form GVL. Other studies show that the process of water addition and isomerization occurs at the initial stage. As shown in Figure 9b, FA undergoes isomerization with water molecules to form intermediate I, which subsequently transforms into GVL through a series of hydrogenation, cyclization, dehydration, hydrogenation, and other reactions.

In recent years, supported catalysts have been increasingly used in GVL production. As a kind of molecular sieve, zeolite has an abundant pore structure and is widely used as a support for GVL production. Bui et al. first used zeolite possessing both Brønsted and Lewis acid sites to synthesize GVL from furfural. With 2-butanol as a hydrogen donor, the GVL yield was close to 80% after a reaction of 24 h at 393 K. They emphasized that the synergy of Brønsted and Lewis was extremely important to the production of GVL [158]. Similarly, Li et al. also emphasized the impact of the Brønsted acid to Lewis acid ratio on GVL production. They synthesized a series of nano-porous SAPO-34 zeolite supported zirconium phosphate catalysts and controlled the ratio of Lewis acid to Brønsted acid by adjusting the Zr/P ratio. Finally, they achieved the maximum GVL selectivity at an L/B ratio of 3.25 [164].

ZSM-5 has been used in the preparation of GVL due to its excellent mesoporous structure and strong acid resistance. Zhu et al. compared the catalytic activity of various solid acid catalysts in GVL production. The combination of Au/ZrO_2_ and ZSM-5 achieved the highest yield. They emphasized that catalysts with medium-strength acid sites demonstrated higher selectivity for GVL. In the catalytic system, ZSM-5 exhibited moderately strong acid sites and a large number of uniformly loaded Au nanoparticles, which would facilitate the production of GVL. In addition, the abundant mesoporous structure of ZSM-5 facilitates the diffusion of reactants and products and accelerates the reaction rate [159].

MCM-41 also possesses a highly mesoporous structure, enabling it to accommodate numerous acidic sites and provide a favorable environment for molecular diffusion. Gao et al. developed a multifunctional catalyst (Fe_3_O_4_/ZrO_2_@MCM-41) using MCM-41 as a support. The characterization of the acid sites showed that the addition of ZrO_2_ would significantly increase the Lewis acid sites on the surface of MCM-41, which would contribute to GVL conversion. Furthermore, the addition of Fe_3_O_4_ not only imparts the catalyst with strong magnetism but also adjusts the acidity of the catalyst to promote GVL production [165].

### 7.6. Conversion of Furfural to Other Chemicals

Furfural hydrogenation is the most common reaction for upgrading furfural to multifunctional compounds. The focus of this paper is primarily focused on the hydrogenation of furfural. However, from a chemical perspective, furfural possesses the potential to be transformed into various other chemicals, in addition to hydrogenation reactions, common reactions that can occur with aldehyde groups, including reductive amination, oxidation to carboxylic acid, decarbonization, etc. Furan rings can also undergo ring opening, alkylation, oxidation, and other reactions [183]. In this section, the amination and oxidation pathways of furfural will be briefly introduced.

Furfurylamine is one of the important downstream chemicals of furfural, which can be synthesized through the reduction amination reaction of furfural under the action of metal catalysts. For example, Maya et al. employed Rh/Al_2_O_3_ as a catalyst, and ammonia and molecular hydrogen as amine sources and reducing agents, respectively, to achieve one-pot conversion of furfurylamine [184]. They highlighted that the main pathway for the formation of furfurylamine is the subsequent amination and reduction reactions in ammonia and hydrogen environments. The conversion of Schiff base-type intermediates into furfurylamine and secondary amine represents a secondary pathway. However, the method of producing furfurylamine using metal catalysts is not appealing due to the severe reaction conditions and the formation of unwanted secondary or tertiary amines. Consequently, researchers have endeavored to combine the specificity of enzymes to efficiently produce furfurylamine from furfural. Di et al. combined a sulfonated Sn-PL catalyst with ω-Transaminase to achieve efficient conversion of biomass to furfurylamine [185]. In this process, the Sn-PL catalyst converted the biomass to furfural, while ω-Transaminase enabled the directed production of furfurylamine. This study provides an important reference for the industrial production of furfurylamine.

Purified furoic acid, an essential downstream derivative of furfural, is extensively employed in the food industry, pharmaceutical synthesis, and other related fields [186]. At present, the production of furoic acid mainly comes from the oxidation of furfural. Biocatalysts, such as *Escherichia coli*, are often employed in this reaction process. Wang et al. used recombinant *Escherichia coli* HMFOMUT as a biocatalyst to facilitate the complete oxidation of furfural to furoic acid in an aqueous reaction medium [187]. In addition, the biocatalyst exhibited remarkable tolerance towards furan compounds. When 5-hydroxymethylfurfural was used as the substrate, the yield of 5-hydroxymethyl-2-furancarboxylic acid could also reach 96.9%. Further exploration revealed that the combination of metal catalysts and biocatalysts could facilitate the efficient conversion of biomass raw materials to furoic acid. Liu et al. prepared a metal catalyst, Sn-FS-RH, for the conversion of C5 sugars in corn cobs to furfural. Then, under the action of biocatalyst, the complete conversion of furfural to furoic acid was achieved [188].

## 8. Summary and Future Prospects

### 8.1. Summary and Existing Issues

As a C5 platform molecule, furfural has great application in various fields such as fuel, solvent, cosmetics, medicine, and others. Over the past few decades, extensive research has been conducted on catalytic production and the high-value utilization of furfural.

As shown in Figure 10a, the application of mineral acids opens the door for the conversion of biomass to furfural, but the issues of “three wastes” and equipment corrosion restrict large-scale production. With the green concept gaining popularity, researchers have explored alternative catalysts such as organic acids, inorganic salts, and ionic liquids for furfural production with promising results (Figure 10b), which can be attributed to effective contact between homogeneous catalysts and reactants. It is also because of this reason that homogeneous catalysts suffer from the drawbacks of separation and reuse, which can be perfectly resolved by heterogeneous catalysts. However, the furfural yield remains relatively low (Figure 10b). In addition, the two-phase solvent system improves furfural production by facilitating its transfer to a purified medium, reducing byproduct formation. Nevertheless, the separation of the target product and the recovery of the solvent pose significant challenges. 

For the upgrading of furfural, we emphasize the utilization of heterogeneous catalysts because we believe this to be a future trend. Therefore, it is an excellent research direction to design heterogeneous catalysts with designable structures. On the other hand, the selection of hydrogen donors plays a pivotal role in furfural upgrading. CTH emerged as a substitute for catalytic hydrogenation because of its relatively safe and resource-conserving catalytic environment. However, the hydrogen donors choice is relatively simple at present (as shown in Table 3, 2-propanol is the most commonly used hydrogen donor). In addition, catalytic furfural upgrading also has a relatively low yield (Figure 10d), particularly for pentanediol due to its numerous isomers and unclear catalytic mechanism.

### 8.2. Future Prospects

In order to solve the above problems and realize the industrial production of furfural, more attention should be devoted to the following future research directions:Strengthen basic theory research

Clarifying the mechanism of chemical reactions is crucial for achieving targeted substance transformations. However, the study of reaction mechanisms in the furfural industry remains challenging due to various factors such as xylose isomerization to furfural, hydrogen proton attack sites during THFL formation, and differences between 1,2-PDO and 1,5-PDO formation processes. Therefore, further experiments are needed to identify the reaction active site and investigate both the target product and byproduct reaction mechanisms. In this process, advanced technology and software can aid in a more comprehensive understanding. For example, in situ infrared spectrometers can identify reaction intermediates, while DFT calculations can provide information on the ground-state energy and surface species properties of the system.

2.Develop new catalysts

Catalysts with excellent performance play a crucial role in achieving high-value utilization of furfural. At present, there are some problems with the catalysts used in the furfural industry. For example, the recovery of homogeneous catalysts poses a challenge and their reusability is poor. Additionally, limited contact between heterogeneous catalysts and reactants can impede catalytic efficiency. Based on this, the key to solving the above problems lies in designing a solid acid catalyst with sufficient diffusion space and uniform active sites. For instance, covalent organic frameworks (COFs) seem to be an excellent candidate for meeting the above conditions, which are crystalline nanomaterials with high designability that can integrate organic units into well-defined two- and three-dimensional polymers. Their exceptional stability, large specific surface area, and facile graft modification render them ideal materials for the catalytic production of furfural. More importantly, the internal pore structure of COF materials can be pre-designed, which will promote the diffusion of reactants and facilitate the directional conversion of furfural. Recently, we synthesized acid-functionalized COFs using monomers containing phenolic hydroxyl groups and triazine rings through an in-situ synthesis strategy. These COFs were applied in the catalytic conversion of xylose to furfural, achieving a furfural yield of 86.7% (Figure 11a). Notably, the catalyst maintained a consistent furfural yield after multiple reuses. Mechanism studies have shown (Figure 11b) that the excellent performance of the catalyst was closely related to the synergistic effect of the triazine ring and phenolic hydroxyl group [189]. It can be seen that functionalized COFs have broad development prospects in the study of biomass conversion for furfural production. In addition, metal-organic frameworks (MOFs) hold significant promise in catalysis, attributed to their porosity, high specific surface area, tunable pore size and functionalities, and efficient catalytic properties. These qualities position MOFs as a research hotspot and a viable candidate in catalytic applications. Currently, the use of MOFs as catalysts for furfural production remains underexplored. However, with ongoing research and technological advances, MOFs are expected to play a crucial role in furfural synthesis, driven by their unique properties and the increasing interest in sustainable, biomass-derived chemical production.

3.Construct green solvent systems

Suitable solvents will help to improve the selectivity of target products and reduce production costs. Single-phase solvent systems are easy to operate but may lead to side reactions. In contrast, biphasic solvent systems offer enhanced selectivity for furfural extraction by introducing an organic phase that is incompatible with water, effectively “capturing” furfural. This setup allows furfural to swiftly transfer from the aqueous to the organic phase, minimizing side reactions and thereby increasing yield and purity. Despite these advantages, biphasic systems face several key challenges. Slow mass transfer rates due to interfacial tension and phase transfer limitations can reduce overall reaction efficiency. Additionally, the separation and recovery of furfural from the organic phase remain complex, necessitating more efficient, cost-effective techniques. The use of organic solvents also poses potential environmental risks if not properly managed. Green solvents, such as ionic liquids, supercritical fluids, and DESs, offer promising, sustainable alternatives with advantages in efficiency, renewability, and lower environmental impact for furfural separation and purification. Notably, DES stands out for its recyclability and potential for reuse. More importantly, some of the ions that make up DES contribute to furfural production; for example, the Cl^−^ in choline chloride could contribute to pentose isomerization. Based on this, our team developed a non-waste biorefinery technology using agricultural waste sugarcane bagasse as the raw material and DES (choline chloride/lactic acid) as the solvent. The results showed that under the synergistic catalysis of lactic acid, AlCl_3_, and choline chloride, hemicellulose derived from sugarcane bagasse could be efficiently converted into furfural with a yield of 68.81% (Figure 11c) [25].

4.Pay attention to the separation and purification of furfural

The separation and purification of furfural are critical steps in its production process. Currently, common methods for furfural purification include distillation and crystallization, which are relatively cumbersome and often yield moderate purity. In contrast, extraction methods offer significant advantages by leveraging the affinity between the extractant and furfural to achieve effective extraction and separation. The extraction method is characterized by excellent selectivity and simple operation. However, the selection and recovery of the extractant are pivotal factors influencing the application of this method. Additionally, membrane separation technology, known for its energy efficiency and environmental friendliness, faces challenges in the separation and purification of furfural. Enhancements in membrane material, membrane structure, and operational conditions could improve the separation efficiency and stability of membrane separation technology. Furthermore, biotechnology presents potential application value in furfural separation and purification due to its environmental friendliness and sustainability. By screening and cultivating micro-organisms with specific enzymatic activities, it is possible to harness their conversion capabilities for furfural separation and purification. This approach not only reduces separation costs but also enables green production of furfural.

5.Enhancing the Recoverability and Reusability of Solid Acid Catalysts

Solid acid catalysts possess a notable advantage: they can be easily separated from soluble reaction systems through simple physical methods. Not only is this process efficient but also it significantly reduces environmental pollution and the burden of waste management. However, insufficient catalytic activity can lead to decreased selectivity for target products and the generation of by-products. These by-products can obstruct mass transfer channels and mask active sites, diminishing the catalyst’s performance during recovery and reuse. Moreover, when solid acid catalysts encounter complex substrates, particularly biomass feedstocks, their incomplete degradation poses additional challenges, complicating catalyst recovery. To enhance the stability and recyclability of solid acid catalysts in complex catalytic systems, we propose several strategies. First, optimizing catalyst design by adjusting the distribution of active sites and improving structural stability can significantly improve catalytic efficiency and durability for furfural production. Second, developing efficient recovery technologies, such as magnetic modification for rapid separation, will facilitate effective catalyst recovery. Third, implementing secondary treatment methods, including calcination in an inert atmosphere or immersion in organic solvents, can effectively remove contaminants from the catalyst surface, enabling regeneration and recycling. Furthermore, incorporating smart materials, like self-cleaning surfaces or responsive polymer coatings, enables dynamic control of catalytic activity and streamlines subsequent recovery processes. These combined approaches are expected to promote the sustainable utilization of solid acid catalysts.

6.Expand the range of hydrogen donors

The hydrogenation reaction is a key step in furfural upgrading, and the use of environmentally friendly hydrogen donors can greatly enhance the overall sustainability of the CTH process. Currently, 2-propanol is the most commonly used hydrogen donor but with relatively limited selectivity (Table 3). Therefore, the future direction of development should be focused on expanding the types of hydrogen donors. The basic principles of expansion should adhere to green and sustainable practices, with no by-products and easy separability. Renewable hydrogen donors such as ethanol and formic acid seem to be excellent substitutes that meet the above conditions.

7.Utilize emerging technologies

Energy serves as the propelling force behind the sustained growth of the furfural industry. At present, furfural production and upgrading consumes a lot of energy, resulting in potential energy waste during heating and mixing processes. The development and utilization of solar energy offer solutions to energy challenges. However, the integration of photocatalytic technology with traditional biomass refining processes should be considered prior to industrialization.

8.Optimize the industrial structure

As a platform compound with great development potential, furfural production is not solely confined to laboratory-scale exploration; several industrial processes for furfural production have been reported. However, the challenge lies in their efficiency and environmental impact. Therefore, in order to further optimize the industrial structure of furfural, a comprehensive consideration of the economic and environmental benefits of the furfural production process is essential. First, the development of a series catalytic system should be vigorously pursued. Compared to the batch reaction system, the series catalytic system boasts advantages such as uninterrupted reaction and high production efficiency, holding the potential to replace the batch process. Secondly, the scale of response should be expanded. In light of potential issues within this process, such as blocked substrate mass transfer, uneven heat distribution, and challenging solvent recovery, it is advocated to design and optimize large-scale reactors through a comprehensive examination of numerical simulation, mass transfer model, reaction kinetics, and solvent characteristics. Finally, upstream and downstream technologies should be enhanced. The preservation and pretreatment of biomass raw materials, the synthesis of catalysts, the storage and transportation of furfural, and the utilization of by-products, all contribute to the economic benefits of the furfural industry. To solve this problem, we can pay attention to the high-value chemicals that may emerge during the furfural production process, because these chemicals can not only generate new market opportunities, but also reduce the overall cost of biorefinery.

As a promising alternative to fossil fuels, furfural has infinite possibilities for future development and can provide impetus for the sustainable development of modern society. However, challenges in production and upgrading processes have hindered the industrialization of furfural, creating a black box effect. Therefore, it is still necessary for researchers to devote their efforts to in-depth research. We trust that our review will offer valuable guidance to researchers and contribute collaboratively to the robust and sustainable growth of the furfural industry.

## Figures and Tables

**Figure 1 ijms-25-11992-f001:**
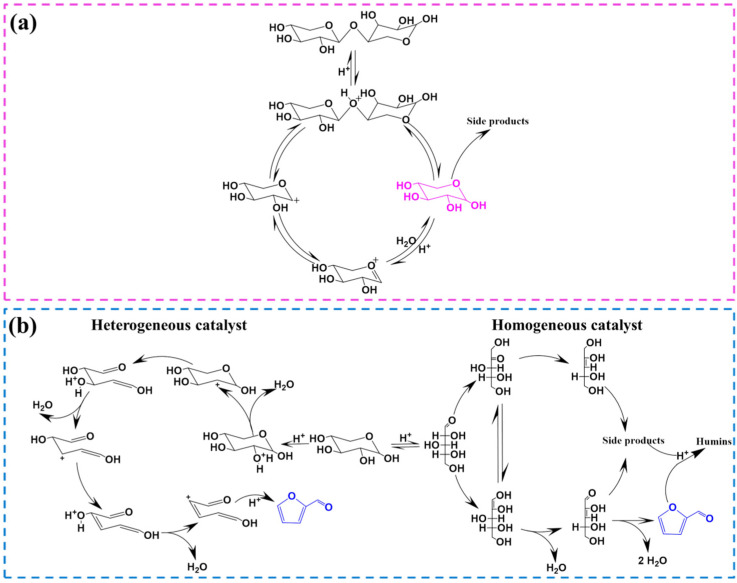
Mechanism of furfural production with acid catalysts. (**a**) Hydrolysis of biomass to pentose (**b**) Dehydration of pentose to form furfural.

**Figure 2 ijms-25-11992-f002:**
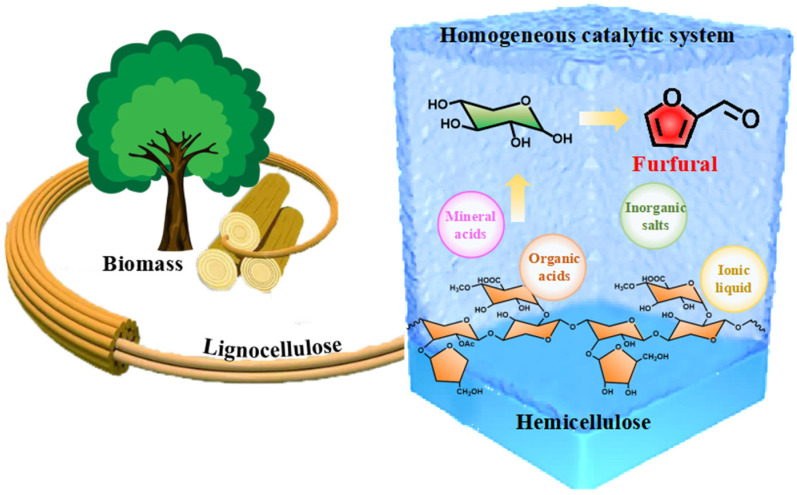
Production process of furfural in a homogeneous system. The yellow arrow represents the production pathway of furfural.

**Figure 3 ijms-25-11992-f003:**
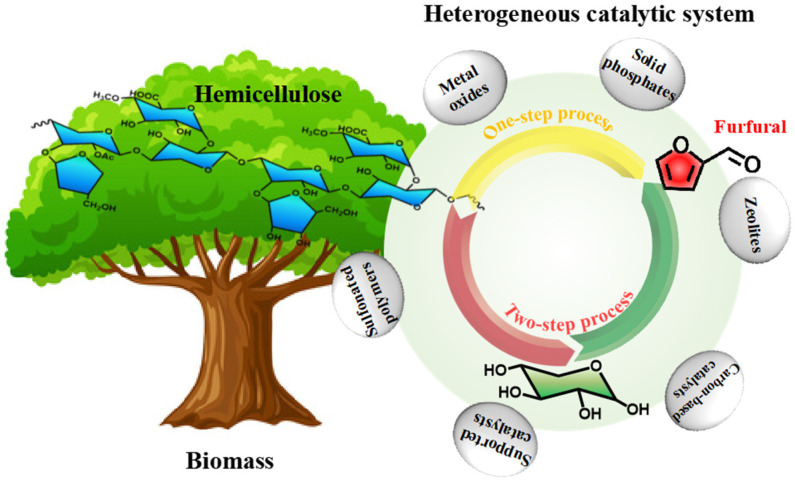
Production process of furfural in a heterogeneous system.

**Figure 4 ijms-25-11992-f004:**
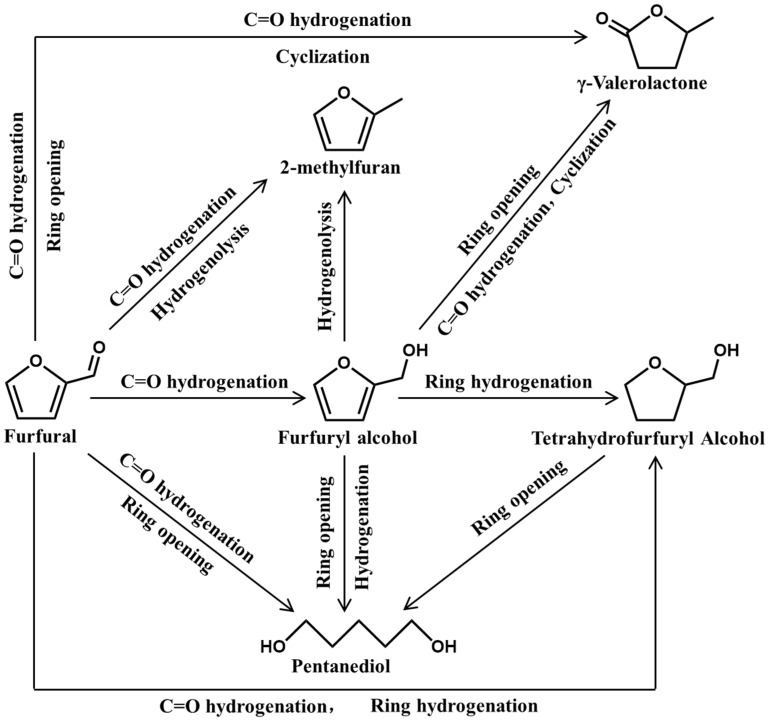
Catalytic upgrading of furfural to C5 chemicals.

**Figure 5 ijms-25-11992-f005:**
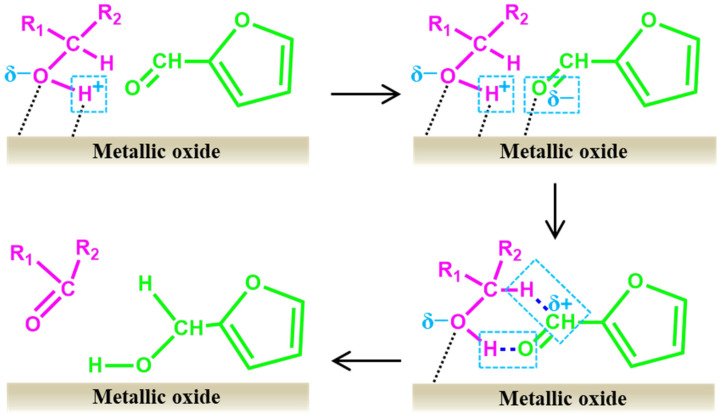
Catalytic mechanism of metal oxide for the upgrading of furfural to FA. The blue box represents the main reaction sites.

**Figure 6 ijms-25-11992-f006:**
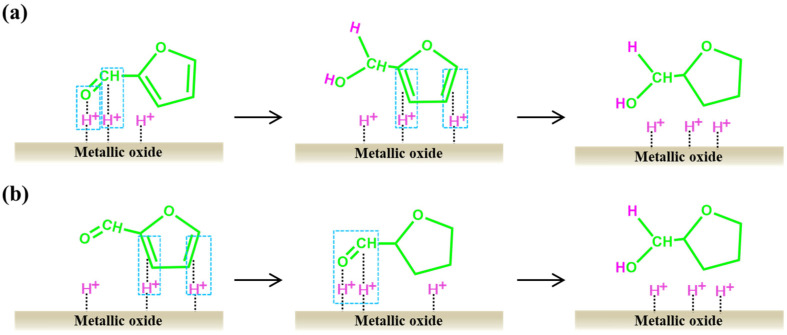
Catalytic mechanism of metal oxide catalyzing furfural upgrading to THFL. (**a**) Preferential hydrogenation of the C=O bond for THFL preparation. (**b**) Preferential hydrogenation of the furan ring for THFL preparation.

**Figure 7 ijms-25-11992-f007:**
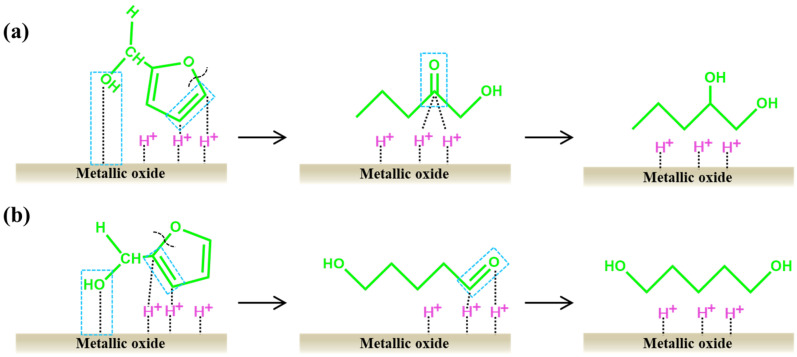
Catalytic mechanism of metal oxide catalyzing furfural upgrading to PDO. (**a**) The C5-O1 bond cleavage leading to 1,5-PDO formation. (**b**) The C2-O1 bond cleavage resulting in 1,5-PDO formation.

**Figure 8 ijms-25-11992-f008:**
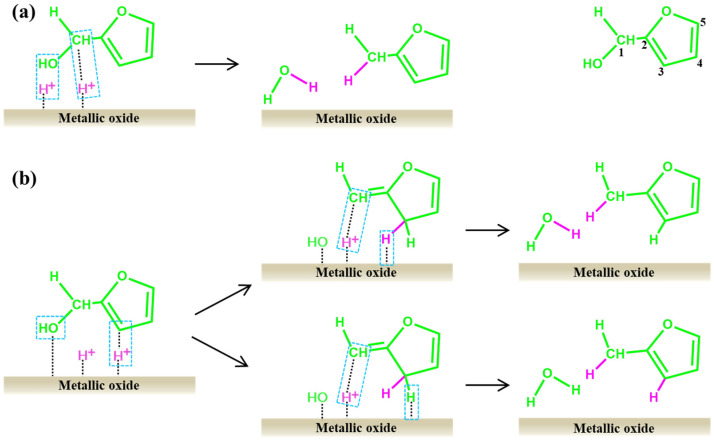
Catalytic mechanism of metal oxide in the upgrading of furfural to MF. (**a**) Cleavage of the C-OH bond leading to the formation of MF. (**b**) Activation of aromatic ring resulting in the formation of MF. The numbers indicate the serial numbers of carbon atoms in the compound, while the blue box highlights the primary reaction site.

**Figure 9 ijms-25-11992-f009:**
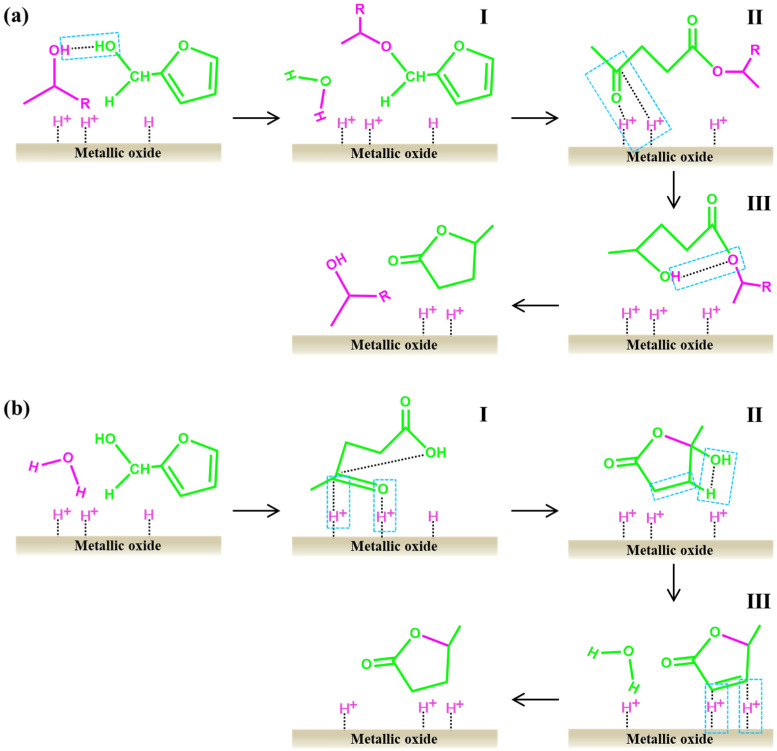
Catalytic mechanism of metal oxide for the upgrading of furfural to GVL. (**a**) Initiate the reaction to form GVL by eliminating water. (**b**) Initiate the reaction to form GVL through the addition of water and an isomerization process. I–III represent reaction intermediates.

**Figure 10 ijms-25-11992-f010:**
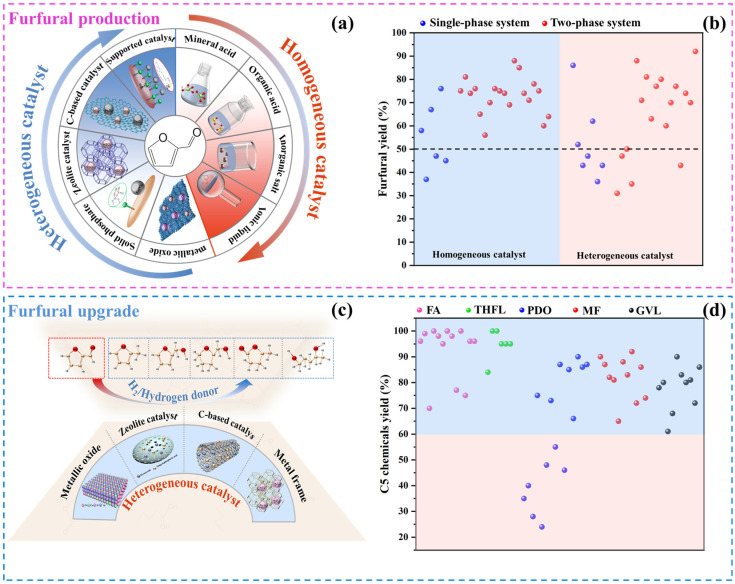
(**a**) The main catalysts used in furfural production, with color depth representing the time sequence used for furfural production. (**b**) Comparison of furfural yield between homogeneous and heterogeneous catalysts mentioned in this paper. (**c**) Commonly used heterogeneous catalysts for the conversion of furfural to C5 compounds. (**d**) Comparison of C5 compound yield mentioned in this paper.

**Figure 11 ijms-25-11992-f011:**
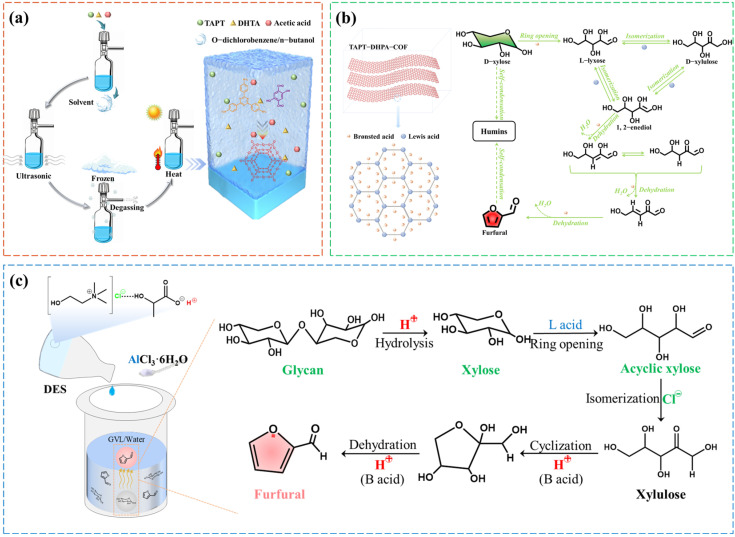
(**a**) Schematic illustration of the synthesis pathway for the functionalized COF (TAPT−DHPA−COF); (**b**) proposed mechanism for the catalytic production of furfural from D−xylose using TAPT−DHPA−COF; (**c**) mechanism of AlCl₃−catalyzed xylan conversion into furfural in a deep eutectic solvent (DES) system. The arrow represents the production pathway of furfural.

**Table 1 ijms-25-11992-t001:** The catalytic production of furfural from biomass using homogeneous catalysts.

Catalyst	Solvent	Reaction Conditions	Substrate Loading	Yield	Selectivity	Ref.
Mineral acids						
H_2_SO_4_, 50 mM	NaCl, 3.5 wt%	200 °C, 10 min	Xylose, 35 mM	75%	83%	[38]
HCl, 50 mM	NaCl, 5.0 wt%	200 °C, 8 min	Xylose, 35 mM	81%	90%	[38]
HCl, pH = 1.12	H_2_O	180 °C, 20 min	Xylan, 1 wt%	34%	-	[39]
H_2_SO_4_, pH = 1.12	H_2_O	180 °C, 20 min	Xylan, 1 wt%	29%	-	[39]
HNO_3_, pH = 1.12	H_2_O	180 °C, 20 min	Xylan, 1 wt%	29%	-	[39]
H_3_PO_4_, pH = 1.12	H_2_O	180 °C, 20 min	Xylan, 1 wt%	26%	-	[39]
H_2_SO_4_, 0.25 wt%	H_2_O	240 °C	Wood chip pre-hydrolysate	76%	80%	[40]
H_2_SO_3_, 0.25 wt%	H_2_O	240 °C	Wood chip pre-hydrolysate	54%	54%	[40]
H_2_SO_4_, 50 mM	H_2_O/MIBK (*v*/*v* = 1/2)	170 °C, 20 min Microwave	Pentose-rich corn stover hydrolyzate, 8 wt%	80%	86%	[41]
Organic acids						
Maleic acid, 0.25 M	H_2_O	200 °C, 28 min Microwave	Xylose, 1 wt%	67%	67%	[42]
Formic acid, 2 wt%	H_2_O/o-Nitrotoluene (*v*/*v* = 1/3)	190 °C, 75 min	Xylose, 8 wt%	74%	86%	[43]
Formic acid, 62.5 g/L, betaine, 17.5 g/L	H_2_O-CPME (*v*/*v* = 1/3)	170 °C, 60 min Microwave	Xylose, 37.5 g/L	76%	76%	[44]
Formic acid, 62.5 g/L, betaine, 17.5 g/L	H_2_O-CPME (*v*/*v* = 1/3)	170 °C, 60 min Microwave	Xylan, 37.5 g/L	80%	80%	[44]
Inorganic salts						
CrCl_3_ 6 mM, HCl 0.1 M	H_2_O-Toluene (*v*/*v* = 1/1)	140 °C, 120 min	Xylose, 1 wt%	76%	80%	[45]
AlCl_3_·6H_2_O, 6.025 g/L, NaCl, 21.875 g/L	H_2_O-THF (*v*/*v* = 1/3)	140 °C, 45 min Microwave	Xylose, 9.375 g/L	75%	75%	[46]
AlCl_3_·6H_2_O, 6.025 g/L, NaCl, 21.875 g/L	H_2_O-THF (*v*/*v* = 1/3)	160 °C, 60 min Microwave	Corn stover, 12.5 g/L	55%	77%	[46]
AlCl_3_·6H_2_O, 6.025 g/L, NaCl, 21.875 g/L	H_2_O-THF (*v*/*v* = 1/3)	160 °C, 60 min Microwave	Switchgrass, 12.5 g/L	56%	79%	[46]
AlCl_3_·6H_2_O, 6.025 g/L, NaCl, 21.875 g/L	H_2_O-THF (*v*/*v* = 1/3)	160 °C, 60 min Microwave	Poplar gave, 12.5 g/L	64%	79%	[46]
FeCl_3_ 10 mol%, NaCl 100 mol%	H_2_O-CPME (*v*/*v* = 1/3)	170 °C, 20 min Microwave	Xylose, 1.25 mmol	74%	74%	[47]
NaHSO_4_, 3.31 wt%	H_2_O-THF (*v*/*v* = 1/10)	190 °C, 90 min	Corncob, 11.1 wt%	54%	54%	[48]
NaHSO_4_, 3.31 wt%	H_2_O-THF (*v*/*v* = 1/10)	190 °C, 90 min	Bagasse, 11.1 wt%	61%	61%	[48]
Al_2_(SO_4_)_3_ 10 mol%	H_2_O-GVL (*v*/*v* = 1/4)	130 °C, 30 min	Xylan, 2 mmol	88%	99%	[49]
Ionic liquids						
[C_4_SO_3_Hpy][BF_4_], 0.1 g	H_2_O-THF (*m*/*m* = 1/2)	180 °C, 60 min Microwave	Xylose, 0.1 g	85%	89%	[50]
[bmim]HSO_4_, 1 g	Toluene, 4.4 g	140 °C, 240 min	Xylose, 0.1 g	74%	74%	[51]
[bmim]HSO_4_, 0.3 g	H_2_O-Toluene (*m*/*m* = 1/4.4)	140 °C, 360 min	Xylose, 0.1 g	71%	72%	[52]
[bmim]HSO_4_, 0.3 g	H_2_O-Toluene (*m*/*m* = 1/4.4)	140 °C, 360 min	Hemicellulose, 0.1 g	62%	-	[52]
[C_3_SO_3_Hmim]Cl, 0.1 M	H_2_O-GVL (*v*/*v* = 1/19)	140 °C, 180 min	Xylose, 30 g/L	78%	82%	[53]
[bmim]AlCl_4_, 0.1 M	H_2_O-Butanone (*v*/*v* = 1/4)	140 °C, 90 min	Xylose, 30 g/L	75%	75%	[54]
[bmim]Cl/AlCl_3_, 0.1 M	H_2_O-Butanone (*v*/*v* = 1/4)	140 °C, 30 min Microwave	Arabinose, 30 g/L	60%	60%	[55]

**Table 2 ijms-25-11992-t002:** Catalytic conversion of biomass to furfural using heterogeneous catalysts.

Catalyst	Solvent	Reaction Conditions	Substrate Loading	Yield	Selectivity	Ref.
Metal oxides						
TiO_2_-ZrO_2_ (1:1), 10 wt%	H_2_O	300 °C, 5 min	Corncob, 10 wt%	10%	42%	[74]
SO_4_^2−^/ZrO_2_-TiO_2_ (7:3), 50 wt%	H_2_O-Butanol (*v*/*v* = 1/1)	170 °C, 120 min	Xylose, 6 wt%	47%	48%	[75]
Nb_2_O_5_, 10 wt%	H_2_O-Toluene (*v*/*v* = 3/7)	170 °C, 90 min	D-xylose, 3 wt%	50%	56%	[76]
Zn-CuO	H_2_O	150 °C, 12 h	Xylose	86%	89%	[77]
Solid phosphates						
ZrP, 0.25 wt%	H_2_O	170 °C, 120 min	Xylose, 1 wt%	52%	54%	[78]
NbP	H_2_O	170 °C, 30 min microwave	Corn straw	23%	-	[79]
CrPO_4_, 0.375 wt%	NaCl/H_2_O/THF (*v*/*v* = 1/3)	180 °C, 90 min	Wheat straw, 2.5 wt%	67%	68%	[80]
CrPO_4_, 0.375 wt%	NaCl/H_2_O/THF (*v*/*v* = 1/3)	160 °C, 60 min	Xylose, 2.5 wt%	88%	89%	[80]
HfP/SiO_2_, 5 wt%	NaCl/H_2_O/THF (*v*/*v* = 1/4)	180 °C, 60 min	Xylan, 10 wt%	85%	-	[81]
Zeolites						
H-MCM-22, 2 wt%	H_2_O-Toluene (*v*/*v* = 3/7)	170 °C, 16 h	Xylose, 3 wt%	71%	72%	[82]
H-mordenite, 4 wt%	GVL (H_2_O 10 wt%)	175 °C, 120 min	Xylose, 2 wt%	81%	81%	[83]
H-mordenite, 4 wt%	GVL (H_2_O 10 wt%)	175 °C, 280 min	Glucose, 0.5 wt%	33%	39%	[83]
HZSM-5, 1.8 wt%	GBL (H_2_O 8.6 wt%)	150 °C, 60 min	Fructose, 5.3 wt%	28%	28%	[84]
HZSM-5, 1.8 wt%	GBL (H_2_O 8.6 wt%)	150 °C, 60 min	Xylose, 4.4 wt%	63%	64%	[84]
SAPO-18, 2 wt%	H_2_O-GVL (*v*/*v* = 1/4)	205 °C, 40 min	Corn stover, 2 wt%	95%	95%	[85]
HSO_3_-ZSM-5, 30 wt%	H_2_O-THF (*v*/*v* = 1/3)	160 °C, 5 h	Corncob, 10 wt%	89%	-	[86]
Cr-deAl-Y, 0.4 wt%	NaCl/H_2_O-n-Butanol (*v*/*v* = 2/3)	180°C, 30 min	Xylose, 0.8 wt%	78%	78%	[87]
Carbon-based catalysts						
SGO, 0.3 wt%	H_2_O	200 °C, 35 min	Xylose, 3 wt%	62%	75%	[88]
SCC, 0.3 wt%	NaCl 13.3 wt %, H_2_O-DCM (*v*/*v* = 1/3)	170 °C, 60 min	Xylose, 5 wt%	81%	83%	[89]
S-MWCNTs, 0.67 wt%	H_2_O	170 °C, 180 min	D-xylose, 3 wt%	36%	57%	[90]
S-RFC, 1.6 wt%	GVL	170 °C, 15 min	Xylose, 2.5 wt%	80%	81%	[91]
S-RFC, 1.9 wt%	GVL	200 °C, 25 min	Corn straw, 2.5 wt%	69%	-	[91]
HSO_3_-C, 0.375 wt%	H_2_O-CPME (*v*/*v* = 1/3)	190 °C, 60 min microwave	D-xylose, 3.75 wt%	60%	62%	[92]
HSO_3_-C, 0.5 wt%	H_2_O-CPME (*v*/*v* = 1/3)	190 °C, 80 min microwave	Xylan, 1.25 wt%	42%	-	[92]
Supported catalysts						
Starbon^®^450-SO_3_H, 0.7 wt%	H_2_O-CPME (*v*/*v* = 1/3)	200 °C, 60 min	Xylose, 2.8 wt%	70%	73%	[93]
SC-GCa-800, 1.0 wt%	GVL	140 °C, 40 min	Xylose, 2.0 wt%	77%	77%	[94]
PSZ-MCM-41, 2.0 wt%	H_2_O-Toluene (*v*/*v* = 3/7)	160 °C, 240 min	Xylose, 3.0 wt%	43%	45%	[95]
SBA-15-SO_3_H, 2.0 wt%	H_2_O-Toluene (*v*/*v* = 3/7)	160 °C, 240 min	Xylose, 3.0 wt%	70%	74%	[96]
SO_3_H-KIT-6, 25 wt%	H_2_O-Toluene (*v*/*v* = 1/1)	170 °C, 120 min	Xylose, 2.0 wt%	92%	94%	[97]
Sulfonated polymers						
PTSA-POM, 1.8 wt%	H_2_O-GVL (*v*/*v* = 1/10)	170 °C, 10 min	Xylose, 2.4 wt%	80%	81%	[98]
PTSA-POM, 1.2 wt%	H_2_O-GVL (*v*/*v* = 1/10)	170 °C, 10 min	Xylan, 2.4 wt%	69%	-	[98]
SSP, 10 wt%	H_2_O (NaCl, 1.5 mmol)-CPME (*v*/*v* = 3/7)	190 °C, 40 min Microwave	D-xylose, 1 mmol	69%	70%	[99]
Nafion NR50, 2 pellets	H_2_O (NaCl, 2.4 wt%)-CPME (*v*/*v* = 1/3)	170 °C, 40 min Microwave	D-xylose, 1 mmol	80%	81%	[100]
Nafion NR50, 2 pellets	H_2_O (NaCl, 2.4 wt%)-CPME (*v*/*v* = 1/3)	170 °C, 60 min Microwave	Xylan, 1 mmol	55%	-	[100]

**Table 3 ijms-25-11992-t003:** Catalytic upgrade of furfural to FA, THFL, PDO, MF, and GVL.

Catalyst	Solvent	Hydrogen Donor	Reaction Conditions	Substrate Loading	Yield	Selectivity	Ref.
Furfuryl alcohol							
Cu/MgO, 1 g	-	H_2_	180 °C, 300 min	Furfural, 1.2 mL/h	96%	98%	[119]
Ni-Co-Al (Ni:Co:Al = 1.1:0.8:1)	-	H_2_	155 °C, 60 min	Furfural, 5.5 mmol/h	70%	72%	[120]
Ni-Fe-B, 2.5 wt%	Alcohol, 30 mL	H_2_, 1 MPa	100 °C, 240 min	Furfural, 10 mL	100%	100%	[121]
5% Pd–5% Cu/MgO, 0.5 wt%	H_2_O	H_2_, 0.6 MPa	110 °C, 80 min	Furfural, 6.0 wt%	99%	99%	[122]
Ru/Zr-MOFs, 1.0 wt%	H_2_O, 9.9 mL	H_2_, 0.5 MPa	20 °C, 240 min	Furfural, 0.1 mL	95%	-	[123]
ZrO_2_, 2.6 wt%	2-butanol, 15 mL	2-butanol	180 °C, 150 min	Furfural, 6.7 wt%	100%	100%	[124]
Cu-Pd/C, 0.2 wt%	1,4-dioxane, 20 mL	Formic acid	170 °C, 180 min	Furfural, 1.5 wt%	98%	98%	[125]
Pd/NPC, 1.2 wt%	2-butanol, 5 mL	2-butanol	120 °C, 600 min	Furfural, 1.9 wt%	77%	84%	[126]
Co-N-C-700, 0.7 wt%	1,4-dioxane, 3.5 mL	Formic acid	150 °C, 300 min	Furfural, 2.0 wt%	100%	100%	[127]
Fe_3_O_4_/C, 0.5 wt%	Isopropanol, 10 mL	Isopropanol	200 °C, 240 min	Furfural, 1.9 wt%	75%	99%	[128]
Mn_3_O_4_, 0.35 wt%	Isopropanol, 7 mL	Isopropanol	220 °C, 240 min	Furfural, 0.5 wt%	96%	96%	[129]
Tetrahydrofurfuryl alcohol							
Cu-Ni/CNT, 2.0 wt%	Ethanol, 5 mL	H_2_, 4 MPa	130 °C, 10 h	Furfural, 0.5 mL	84%	85%	[130]
Ni/C-500, 0.6 wt%	2-propanol, 5 mL	H_2_, 3 MPa	120 °C, 120 min	Furfural, 0.6 wt%	100%	100%	[131]
Pd-HAP, 0.3 wt%	2-propanol,10 mL	H_2_, 1 MPa	40 °C, 180 min	Furfural, 1.0 wt%	100%	100%	[132]
CuNi/MgAlO, 0.25 wt%	Ethanol, 20 mL	H_2_, 4 MPa	150 °C, 180 min	Furfural, 2.4 wt%	95%	95%	[133]
Pd-Pt/TiO_2_, 0.97 wt%	2-propanol, 15 mL	H_2_, 0.3 MPa	30 °C, 120 min	Furfural, 2.9 wt%	95%	95%	[134]
Ni@C@CNT, 0.6 wt%	Ethanol, 100 mL	H_2_, 4 MPa	120 °C, 240 min	Furfural, 10 wt%	95%	96%	[135]
1,5-Pentanediol							
Pt/Co_2_AlO_4_, 0.4 wt%	H_2_O	H_2_, 1.5 MPa	140 °C, 24 h	Furfural, 0.8 wt%	35%	35%	[136]
CoAlO-r700, 0.1 wt%	2-propanol, 40 mL	H_2_, 4 MPa	150 °C, 480 min	Furfural, 1.0 wt%	40%	43%	[137]
Pt/HT, 3.3 wt%	2-propanol, 3 mL	H_2_, 3 MPa	60 °C, 480 min	Furfural, 3.2 wt%	28%	28%	[138]
Pt@Al_2_O_3_, 0.125 wt%	H_2_O,10 mL	H_2_, 0.45 MPa	45 °C, 480 min	Furfural, 0.4 wt%	75%	75%	[139]
Ni-CoOx-Al_2_O_3_, 0.25 wt%	Ethanol, 40 mL	H_2_, 3 MPa	160 °C, 360 min	Furfural, 0.5 wt%	48%	48%	[140]
1,2-Pentanediol							
Pt/HT, 3.3 wt%	2-propanol,3 mL	H_2_, 3 MPa	150 °C, 480 min	Furfural, 3.2 wt%	73%	73%	[138]
2Pd2AuTiSBA, 0.26% wt%	IPA, 95 mL	H_2_, 3.45 MPa	160 °C, 300 min	Furfural, 2.6 wt%	55%	59%	[141]
Rh/OMS-2, 1.25 wt%	Methanol, 20 mL	H_2_, 3 MPa	160 °C, 480 min	Furfural, 3.5 wt%	87%	87%	[142]
1Ru-5Sn/ZnO, 1.0 wt%	2-propanol, 10 mL	H_2_ 3.5 MPa	140 °C, 360 min	Furfural, 1.0 mL	85%	85%	[143]
3%Pd/MMT-K10, 0.26 wt%	2-propanol, 95 mL	H_2_, 3.45 MPa	220 °C, 300 min	Furfural, 2.6 wt%	66%	66%	[144]
1,4-Pentanediol							
Ru/CMK-3-R200, 0.57 mol%	H_2_O, 5 mL	CO_2_/H_2_ (3 MPa/1 MPa)	80 °C, 20 h	Furfural, 2.0 wt%	90%	90%	[145]
Ru-6.3FeOx/AC	H_2_O	H_2_	80 °C		86%	86%	[146]
Ru/SC-SBA-15, 2.0 wt%	H_2_O, 4 mL	H_2_ 1.5 MPa	140 °C, 240 min	Furfural, 4.8 wt%	87%	87%	[147]
2-Methylfuran							
Cu/SiO_2_	H_2_O	H_2_	220 °C, 30 min		90%	90%	[148]
24%Cu-Ps	-	H_2_	200 °C, 90 min		87%	87%	[149]
Cu-Ni/γ-Al_2_O_3_, 5.0 mol%	2-propanol, 20 mL	H_2_, 4 MPa	200 °C, 240 min	Furfural, 21.6 wt%	82%	82%	[150]
PtCo_3_	1-propanol	H_2_	180 °C, 300 min		81%	81%	[151]
10Cu-3Pd/ZrO_2_, 0.86 wt%	2-propanol, 14 mL	2-propanol	220 °C, 240 min	Furfural, 0.69 wt%	65%	66%	[152]
Cu_3_Al-A, 0.6 wt%	Methanol, 15 mL	Methanol	240 °C, 90 min	Furfural, 0.77 wt%	88%	90%	[153]
4%Ru/NiFe_2_O_4_, 0.625 wt%	2-propanol, 24 mL	2-propanol	180 °C, 360 min	Furfural, 1.0 wt%	83%	85%	[154]
Cu/AC, 0.38 wt%	2-propanol, 5 mL	2-propanol	200 °C, 300 min	Furfural, 1.15 wt%	92%	92%	[155]
CuZnAl, 0.6 wt%	Isopropanol, 25 mL	Isopropanol	180 °C, 240 min	Furfural, 2.0 wt%	72%	72%	[156]
CuZnAl, 3.3 wt%	Methanol, 3 mL	Methanol	240 °C, 300 min	Furfural, 3.3 wt%	74%	74%	[157]
γ-Valerolactone							
Al-MFI-ns, 6.7 wt%	H_2_O-2-butanol	2-butanol	120 °C, 48 h	Furfural, 4.36 wt%	78%	90%	[158]
Au/ZrO_2_ 2.0 wt%, ZSM-5 2.0 wt%	2-propanol, 10 g	2-propanol	120 °C, 24 h	Furfural, 0.48 wt%	80%	89%	[159]
Sn-Al-Beta7, 0.8 wt%	2-butanol, 20 mL	2-butanol	180 °C, 24 h	Furfural, 1.0 wt%	61%	61%	[160]
HPW/Zr-Beta	2-propanol, 20 mL	2-propanol	160 °C, 24 h	Furfural, 1.0 wt%	68%	68%	[161]
ZrO_2_-TPA-β zeolite, 0.75 wt%	2-propanol, 20 mL	2-propanol	170 °C, 10 h	Furfural, 1.0 wt%	90%	90%	[162]
ZrO_2_-[Al]MFI-NS 30, 0.15 wt%	2-propanol	2-propanol	170 °C, 36 h	Furfural, 0.38 wt%	83%	83%	[163]
ZPS-1.0, 0.96 wt%	2-propanol, 20 mL	2-propanol	170 °C, 6 h	Furfural, 0.48 wt%	80%	80%	[164]
Fe_3_O_4_/ZrO_2_@MCM-41, 0.4 wt%	Isopropanol, 10 mL	Isopropanol	150 °C, 24 h	Furfural, 0.48 wt%	81%	81%	[165]
Ni-Co-Fe/ZSM-5, 1.2 wt%	Ethanol, 5 mL	H_2_, 4 MPa	150 °C, 14 h	Furfural, 0.77 wt%	86%	86%	[166]

## Data Availability

Data supporting the reported results can be obtained from the authors on request.

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
