# Peer review of "Catalytic Production and Upgrading of Furfural: A Platform Compound"

_ijms, 2024, doi:10.3390/ijms252211992_

Round 1
Reviewer 1 Report
Comments and Suggestions for Authors
This manuscript is an interesting review about the chemistry involved in the production of furfural and, from this platform molecule, to obtain other chemicals (mainly C5 chemicals). As a green bio-based chemical molecule, the interest in furfural is steadily increasing and, subsequently, the topic of the review is appealing. The consulted literature also seems appropriate. From my point of view, the manuscript could be accepted in the International Journal of Molecular Sciences after minor improvements.
1) The authors have made a mistake with the title of section 5, which should refer to heterogeneous catalysts for furfural production from biomass, not homogeneous ones. Please correct this error.
2) In the section 4, related to homogeneous catalysts, authors have revised organic and inorganic acids and ionic liquid but not the role of catalysts made from metal complexes. I recommend completing this section with a subsection to deal with this type of compound.
3) I would suggest rewriting section 3.2 reaction kinetics of furfural production. In this section, the authors seem to have limited themselves to presenting the results of some publications in a somewhat disjointed manner. A more coherent discussion of these studies is required.
4) Please rewrite the sentence at line 67 “the molecular structure of furfural contains aldehydes, ethers and other compounds”.
5) I would like that, in the proposal for the development of new catalysts, in which the authors propose covalent organic frameworks (COFs), metal-organic frameworks (MOFs) would also be discussed, or at least, the advantages that COFs would have over MOFs when the authors are inclined to highlight the former.
Author Response
Reviewer #1:
First, thank you for your positive comments. Following are the responses or discussions to each comment (in black), the deleted parts and the revised parts in manuscript are marked in red and blue, respectively.
Issue 1:
The authors have made a mistake with the title of section 5, which should refer to heterogeneous catalysts for furfural production from biomass, not homogeneous ones. Please correct this error.
Response
Thanks for your comment.
We have revised the title of Section 5. Additionally, we identified a similar issue with the title of Section 4.1, which has also been corrected.
Revision parts:
“5. Homogeneous catalysts for the production of furfural from biomass” was revised to “5. Heterogeneous catalysts for the production of furfural from biomass”
“4.1. Reaction kinetics of furfural production” was revised to “4.1. Mineral acids”
Issue 2:
In the section 4, related to homogeneous catalysts, authors have revised organic and inorganic acids and ionic liquid but not the role of catalysts made from metal complexes. I recommend completing this section with a subsection to deal with this type of compound.
Response
Thanks for your comment.
We have updated the article to include recent advancements in metal complex-catalyzed production of furfural from biomass-derived sugars.
Supplementary parts:
4.5. Metal complexes
Metal complex catalysts are coordination compounds formed by transition metal ions or atoms bonded with organic ligands, imparting catalytic activity [70]. The key advantage of metal complex catalysts lies in their ability to finely tune catalyst properties by modifying the type and structure of metal ions and ligands, thereby achieving high yields and purity of the target product. However, their application in furfural production is relatively limited due to challenges in recyclability.
Ana et al. evaluated the catalytic performance of various metal complexes, specifically including H3PW12O40 (PW), H4SiW12O40 (SiW), and H3PMo12O40 (PMo), for the liquid-phase catalysis of D-xylose to furfural [71]. Using DMSO as the solvent and a reaction temperature of 140 °C, the tungsten-based metal complexes achieved furfural yields comparable to those of H2SO4 and p-toluenesulfonic acid catalysts after 4 h (58–67%), whereas PMo yielded less than half this amount of furfural. The study also highlighted the significant effect of solvent choice on catalytic performance: in DMSO, furfural yields ranked PW > SiW > PMo, while in an aqueous system, the order reversed to PMo > PW > SiW. Geonu et al. compared the catalytic performance of metal complexes, inorganic acids, and ion-exchange resins in furfural production, specifically evaluating PW, H2SO4, and Amberlyst 15 for alginic acid conversion [72]. PW catalyst exhibited the highest catalytic activity, yielding a maximum of 33.8% furfural in a tetrahydrofuran/water co-solvent at 180 °C within 30 min. Based on product yield analysis over time, the authors proposed that furfural production involved the hydrolysis of alginic acid into monomers, followed by decarboxylation and dehydration to generate furfural.
Given the limited catalytic activity of single-metal complexes in furfural preparation, researchers have explored bimetallic complexes to improve yields. Ana et al. synthesized CsxH3-xPW12O40 (CsxPW) based on PW and applied it to the liquid-phase catalysis of xylose to furfural [73]. The results indicated that the bimetallic complex CsxPW exhibited higher catalytic activity than PW, likely due to a synergistic effect between W and Cs. Additionally, they supported CsxPW on silica to obtain a solid acid catalyst, which demonstrated high selectivity for furfural production.
While issues like low yield and limited recyclability constrain the industrial use of metal complex catalysts in furfural production, advantages such as adjustable structure and tunable metal ions make them promising for furfural production.
Issue 3:
I would suggest rewriting section 3.2 reaction kinetics of furfural production. In this section, the authors seem to have limited themselves to presenting the results of some publications in a somewhat disjointed manner. A more coherent discussion of these studies is required.
Response
Thanks for your comment.
We have systematically revisited the reaction kinetics associated with furfural production. Based on the production process, the reaction kinetics are divided into two primary sections for discussion: the hydrolysis of hemicellulose to pentoses and the subsequent dehydration of pentoses to furfural. In analyzing the kinetics of hemicellulose hydrolysis to pentoses, we investigated the substantial influence of acid site distribution within the catalytic system on reaction kinetics. This examination includes both homogeneous and heterogeneous catalysts. Similarly, in exploring the kinetics of pentose dehydration to furfural, we recognized the complexity of this reaction. We thoroughly elucidated the kinetic characteristics by examining two key aspects: the involvement of furfural in side reactions and its absence from these reactions.
Revision parts:
“In this section, the reaction kinetics … and the degradation reaction of furfural cannot be avoided in the production process of furfural.” was revised to
“The construction of kinetic models is essential for understanding and optimizing the production process of target products. In furfural production, the kinetic process primarily involves the acid hydrolysis and dehydration reactions of biomass resources, predominantly hemicellulose.
Firstly, during the acid hydrolysis stage, hemicellulose in biomass is hydrolyzed into monosaccharides such as xylose and arabinose under acidic conditions. The kinetic parameters of the hydrolysis reaction are primarily influenced by the reaction system. In homogeneous catalytic systems, hydrogen ions are uniformly dispersed, and the hydrolysis of hemicellulose is considered a first-order reaction, with the hydrolysis rate of oligosaccharides being faster than that of polysaccharides [30, 31]. For instance, Kamiyama et al. noted that the hydrolysis rate of di-oligosaccharides was about 1.8 times that of penta-oligosaccharides [32]. In contrast, heterogeneous catalytic systems exhibit significant variations in the hydrolysis rate of hemicellulose. The hydrolysis rate is initially slow, accelerates during the middle stage, and then slows down again, eventually stabilizing. This behavior is more consistent with a second-order kinetic model, which is attributed to the changing collision probability between hemicellulose and active sites [33]. In heterogeneous systems, acidic sites are only present on the surface of the catalyst. The initial hydrolysis rate is determined by random collisions between hemicellulose molecules and acid sites [34]. As the reaction progresses, hemicellulose is hydrolyzed into oligosaccharides, increasing the collision probability with acid sites and consequently the hydrolysis rate. In the later stages of the reaction, the observed slowdown in the reaction rate can be explained by internal mass transfer limitations and steric hindrance, as smaller molecular substances may obstruct the pore structure of the heterogeneous catalyst [35].
In the dehydration stage, pentoses such as xylose and arabinose further dehydrate in an acidic environment to form furfural. The kinetic model for furfural formation is primarily related to side reactions involving furfural itself. Assuming that furfural does not participate in side reactions, the kinetic model for furfural formation aligns with that of hemicellulose hydrolysis, conforming to a pseudo-first-order model in homogeneous systems and a second-order model in heterogeneous systems [26, 36]. However, in actual reaction processes, both condensation reactions involving xylose and furfural and degradation reactions of furfural take place. Therefore, a modified kinetic model is necessary to accurately represent the furfural production and consumption process.”
Issue 4:
Please rewrite the sentence at line 67 “the molecular structure of furfural contains aldehydes, ethers and other compounds”.
Response
Thanks for your comment.
The above sentence has been revised and corrected.
Revision parts:
Line 67: “the molecular structure of furfural contains aldehydes, ethers and other compounds” was revised to “the molecular structure of furfural includes a furan ring with an aldehyde group, along with other functional groups”
Issue 5:
I would like that, in the proposal for the development of new catalysts, in which the authors propose covalent organic frameworks (COFs), metal-organic frameworks (MOFs) would also be discussed, or at least, the advantages that COFs would have over MOFs when the authors are inclined to highlight the former.
Response
Thanks for your comment.
Metal-organic frameworks (MOFs) are promising as catalysts due to their high surface area, tunable functional groups, and abundant uncoordinated active sites. Additionally, MOFs hold significant potential in the furfural production. Consequently, we have included MOFs into our future prospects section.
Supplementary parts:
“In addition, Metal-organic frameworks (MOFs) hold significant promise in catalysis, attributed to their porosity, high specific surface area, tunable pore size and functionalities, and efficient catalytic properties. These qualities position MOFs as a research hotspot and a viable candidate in catalytic applications. Currently, the use of MOFs as catalysts for furfural production remains underexplored. However, with ongoing research and technological advances, MOFs are expected to play a crucial role in furfural synthesis, driven by their unique properties and the increasing interest in sustainable, biomass-derived chemical production. ”
Reviewer 2 Report
Comments and Suggestions for Authors
This paper deals with the catalytic production and upgrading of furfural, a platform compound derived from renewable lignocellulosic biomass. This paper is informative because furfural is a biomass-derived compound that has gained traction in recent years as a potential alternative to fossil resources. However, it needs some minor revisions before it can be published.
1. The names of the subheadings are redundant. “4. Homogeneous catalysts for producing furfural from biomass” and ‘5. Homogeneous catalysts for producing furfural from biomass’ have the same name. This should be corrected to avoid duplication.
2. More images are needed to improve the informativeness of the review article. The authors should add images to illustrate the conversion of furfural from biomass by different processing methods or the conversion of furfural to other compounds, either by creating their images or by referring to high-impact studies.
3. improving furfural yield remains a significant challenge, but the technology to “purify and separate the generated furfural” is equally important. These implications should be provided in “8. Summary and Future Prospects”. “8. Summary and Future Prospects” indicates that a dual-phase solvent system should be used, which has already been used for several years. A more novel method (or a different, very modern method) should be presented.
Author Response
Reviewer #2:
First, thank you for your positive comments. Following are the responses or discussions to each comment (in black), the deleted parts and the revised parts in manuscript are marked in red and blue, respectively.
Issue 1:
The names of the subheadings are redundant. “4. Homogeneous catalysts for producing furfural from biomass” and ‘5. Homogeneous catalysts for producing furfural from biomass’ have the same name. This should be corrected to avoid duplication.
Response
Thanks for your comment.
We have revised the title of Section 5. Additionally, we identified a similar issue with the title of Section 4.1, which has also been corrected.
Revision parts:
“5. Homogeneous catalysts for the production of furfural from biomass” was revised to “5. Heterogeneous catalysts for the production of furfural from biomass”
“Reaction kinetics of furfural production” was revised to “Mineral acids”
Issue 2:
More images are needed to improve the informativeness of the review article. The authors should add images to illustrate the conversion of furfural from biomass by different processing methods or the conversion of furfural to other compounds, either by creating their images or by referring to high-impact studies.
Response
Thanks for your comment.
To enhance the information content and readability of this review, we have incorporated several graphical materials into the manuscript. Specifically, we added the following images: “Figure 2: Production process of furfural in homogeneous system”; “Figure 3: Production process of furfural in a heterogeneous system”; “Figure 11: (a) Schematic illustration of the synthesis pathway for the functionalized covalent organic framework (COF) TAPT-DHPA-COF; (b) Proposed mechanism for the catalytic production of furfural from D-xylose using TAPT-DHPA-COF; (c) Mechanism of AlCl₃-catalyzed xylan conversion to furfural in a deep eutectic solvent (DES) system”. Furthermore, we have modified the corresponding text sections referencing these images and renumbered all figures to ensure logical coherence and consistency.
Supplementary parts:
Figure 2. Production process of furfural in homogeneous system.
Figure 3. Production process of furfural in a heterogeneous system.
Figure 11. (a) Schematic illustration of the synthesis pathway for the functionalized COF (TAPT-DHPA-COF); (b) Proposed mechanism for the catalytic production of furfural from D-xylose using TAPT-DHPA-COF; (c) Mechanism of AlCl₃-catalyzed xylan conversion into furfural in a deep eutectic solvent (DES) system.
Revision parts:
“Homogeneous catalysts used in the traditional furfural production process include mineral acids, organic acids, inorganic salts and ionic liquids (Table 1)” was revised to “Figure 2 illustrates the homogeneous catalysts commonly used in the furfural production process, including inorganic acids, organic acids, inorganic salts and ionic liquids. Table 1 summarizes the research progress associated with these catalysts. ”
“Metal oxides, solid phosphates, zeolites, carbon-based catalysts, supported catalysts and sulfonated polymers are among the commonly used types of solid acid catalysts (Table 2).” was revised to “Figure 3 depicts the commonly utilized types of solid acid catalysts, which encompass metal oxides, solid phosphates, zeolites, carbon-based catalysts, supported catalysts, and sulfonated polymers. Table 2 outlines the research progress associated with these catalysts.”
“Recently, we synthesized acidic functionalized COFs using monomers containing phenolic hydroxyl groups and triazine rings through an in-situ synthesis strategy, and applied them in the catalytic conversion of xylose to furfural. The result achieved a furfural yield of 86.7%. In addition, the catalyst had excellent stability. After six repeated uses, there was no significant decrease in the yield of furfural” was revised to “Recently, we synthesized acidic functionalized COFs using monomers containing phenolic hydroxyl groups and triazine rings through an in-situ synthesis strategy. These COFs were applied in the catalytic conversion of xylose to furfural, achieving a furfural yield of 86.7% (Figure 11a). Notably, the catalyst maintained a consistent furfural yield after multiple reuses.. Mechanism studies had shown (Figure 11b) that the excellent performance of the catalyst was closely related to the synergistic effect of the triazine ring and phenolic hydroxyl group.”
“The results showed that under the synergistic catalysis of lactic acid, AlCl3 and choline chloride, hemicellulose derived from sugarcane bagasse could be efficiently converted into furfural with a yield of 68.81%” was revised to “The results showed that under the synergistic catalysis of lactic acid, AlCl3 and choline chloride, hemicellulose derived from sugarcane bagasse could be efficiently converted into furfural with a yield of 68.81% (Figure 11c)”
“Figure 2” was revised to “Figure 4”
“Figure 3” was revised to “Figure 5”
“Figure 4” was revised to “Figure 6”
“Figure 5” was revised to “Figure 7”
“Figure 6” was revised to “Figure 8”
“Figure 7” was revised to “Figure 9”
“Figure 8” was revised to “Figure 10”
Issue 3:
Improving furfural yield remains a significant challenge, but the technology to “purify and separate the generated furfural” is equally important. These implications should be provided in “8. Summary and Future Prospects”. “8. Summary and Future Prospects” indicates that a dual-phase solvent system should be used, which has already been used for several years. A more novel method (or a different, very modern method) should be presented.
Response
Thanks for your comment.
We have revised Section 8, titled “Summary and Future Prospects.” This includes the addition of furfural purification and separation technology, as well as modifications to the solvent system section.
Supplementary parts:
- Pay attention to the separation and purification of furfural
The separation and purification of furfural are critical steps in its production process. Currently, common methods for furfural purification include distillation and crystallization, which are relatively cumbersome and often yield moderate purity. In contrast, extraction methods offer significant advantages by leveraging the affinity between the extractant and furfural to achieve effective extraction and separation. The extraction method is characterized by excellent selectivity and simple operation. However, the selection and recovery of the extractant are pivotal factors influencing the application of this method. Additionally, membrane separation technology, known for its energy efficiency and environmentally friendly, faces challenges in the separation and purification of furfural. Enhancements in membrane material, membrane structure, and operational conditions could improve the separation efficiency and stability of membrane separation technology. Furthermore, biotechnology presents potential application value in furfural separation and purification due to its environmental friendliness and sustainability. By screening and cultivating microorganisms with specific enzymatic activities, it is possible to harness their conversion capabilities for furfural separation and purification. This approach not only reduces separation costs but also enables green production of furfural.
Revision parts:
“Although single-phase solvent systems are easy … Because DES with low toxicity has a greater advantage in recycling.” was revised to “Single-phase solvent systems are easy to operate but may lead to side reactions. In contrast, biphasic solvent systems offer enhanced selectivity for furfural extraction by introducing an organic phase that is incompatible with water, effectively "capturing" furfural. This setup allows furfural to swiftly transfer from the aqueous to the organic phase, minimizing side reactions and thereby increasing yield and purity. Despite these advantages, biphasic systems face several key challenges. Slow mass transfer rates due to interfacial tension and phase transfer limitations can reduce overall reaction efficiency. Additionally, the separation and recovery of furfural from the or-ganic phase remain complex, necessitating more efficient, cost-effective techniques. The use of organic solvents also poses potential environmental risks if not properly managed. Green solvents, such as ionic liquids, supercritical fluids, and DESs, offer promising, sustainable alternatives with advantages in efficiency, renewability, and lower environmental impact for furfural separation and purification. Notably, DES stands out for its recyclability and potential for reuse.”
Reviewer 3 Report
Comments and Suggestions for Authors
The manuscript of Gan et al. reviews the production of furfural via various catalysts and its subsequent conversion into various compounds Although the production and subsequent conversion of furans has been excessively studied already (with numerous catalysts including metal oxides/metal phosphates/MOFs and several (biphasic) solvent system, etc), there are some interesting aspects of this study. Therefore this paper deserves recognition in this field of research. The approach of the authors is well performed, the article is well written and well structured, and therefore, this manuscript merits publication in the International Journal of Molecular Sciences. However, after reading the manuscript I have some major and minor comments which ought to be addressed before publication:
- Lines 61-63: The authors mention the following statement: “Heterogeneous catalysts have an obvious phase interface with the reactants, which makes them easy to be recycled and reused, but they usually require external forces to strengthen the contact with the reactants”. This is not entirely correct. When effective biomass is used and the heterogenous catalyst is mixed with the biomass, recycling of the catalyst can be very tedious. Please elaborate in the text.
- The reviewer does not really understand the statement mentioned in Lines 76-78: “Compared to 5-hydroxymethylfurfural, furfural production needs more attention owing to fewer of biomass feedstock for its synthesis.” Please elaborate.
- Lines 108-109: please add references.
- Line 151: a space is missing.
- Add references for Lines 193-200. Also for lines 208-210. Also for lines 245-249.
- Is reference 39 also applicable to Lines 271-280? Otherwise provide reference(s) for those Lines.
- Also add reference(s) for lines 404-407.
- The authors should really elaborate on the recyclability of solid catalysts for the production of furfural from biomass. For several of the mentioned solid catalysts there is stated that they have “strong” recyclability. However, working with solid biomass and/or the formation of humins this can be not that straightforward. Please elaborate in the text.
- Line 538: space missing.
- Line 591 and 594: what is meant by “workshop”?
- Please provide a reference for Lines 649-650.
- Also for Lines 704-707.
- Also for Lines 725-726.
- Also for Lines 834-847.
- Line 893: remove “.” In gamma-valerolactone.
- Line 900-901: add reference(s).
- The authors mention a couple of times the usage of biphasic systems. Maybe this deserves a separate paragraph and a more in-depth discussion?
- Section 8.2.2: please also address the recyclability issue of solid catalysts.
Author Response
Reviewer #3:
First, thank you for your positive comments. Following are the responses or discussions to each comment (in black), the deleted parts and the revised parts in manuscript are marked in red and blue, respectively.
Issue 1:
Lines 61-63: The authors mention the following statement: “Heterogeneous catalysts have an obvious phase interface with the reactants, which makes them easy to be recycled and reused, but they usually require external forces to strengthen the contact with the reactants”. This is not entirely correct. When effective biomass is used and the heterogenous catalyst is mixed with the biomass, recycling of the catalyst can be very tedious. Please elaborate in the text.
Response
Thanks for your comment.
The above statement assumes the absence of solid biomass residues post-reaction, which introduces certain limitations. Therefore, we have supplemented the initial assumptions. Additionally, to address the difficulty of separating heterogeneous catalysts from solid residues, targeted solutions such as magnetic separation, high-temperature calcination, and organic solvent elution are proposed in Section 8, “Future Prospects”.
Revision parts:
“Heterogeneous catalysts have an obvious phase interface with the reactants, which makes them easy to be recycled and reused, but they usually require external forces to strengthen the contact with the reactants” was revised to “Heterogeneous catalysts have an obvious phase interface with solvents, making them easy to recycle and reuse in the absence of solid biomass residues after the reaction, but they usually require external forces to strengthen the contact with the reactants”
Issue 2:
The reviewer does not really understand the statement mentioned in Lines 76-78: “Compared to 5-hydroxymethylfurfural, furfural production needs more attention owing to fewer of biomass feedstock for its synthesis.” Please elaborate.
Response
Thanks for your comment.
We emphasize that, compared to 5-HMF production, furfural preparation offers clear raw material advantages and cost-efficiency. To enhance clarity, we have rephrased and elaborated on this argument, with supporting citations from relevant literature.
Revision parts:
Lines 76-78: “Compared to 5-hydroxymethylfurfural, furfural production needs more attention owing to fewer of biomass feedstock for its synthesis. Therefore, developing efficient catalytic technologies will contribute to the furfural industry.” was revised to “Compared to 5-hydroxymethylfurfural production, furfural offers significant advantages, including a broad range of raw materials, such as hemicellulose from agricultural and forestry residues, and relatively low production costs [15]. These characteristics give furfural greater potential in the chemical industry, making it a worthy focus for further research and development.”
Issue 3:
Lines 108-109: please add references.
Response
Thank you for your review.
We have supplemented the references.
Supplementary parts:
However, recent studies have found that C6 sugars also have great potential in the furfural production, such as glucose, fructose, etc [9].
Issue 4:
Line 151: a space is missing.
Response
Thanks for your comment.
We have added spaces.
Supplementary parts:
Then, the unsaturated aldehyde undergoes dehydration and cyclization to form furfural under the action of Brønsted acid (Figure 1b) [26]. However, another dehydration mechanism exists when heterogeneous catalysts are used to produce furfural (Figure 1b).
Issue 5:
Add references for Lines 193-200. Also for lines 208-210. Also for lines 245-249.
Response
Thanks for your comment.
We have supplemented the references. Furthermore, to enhance readability, we have also made minor modifications to the content of the article.
Supplementary parts:
Lines 193-200: In heterogeneous systems, acidic sites are only present on the surface of the catalyst. The initial hydrolysis rate is determined by random collisions between hemicellulose molecules and acid sites [34]. As the reaction progresses, hemicellulose is hydrolyzed into oligosaccharides, increasing the collision probability with acid sites and conse-quently the hydrolysis rate. In the later stages of the reaction, the observed slowdown in the reaction rate can be explained by internal mass transfer limitations and steric hindrance, as smaller molecular substances may obstruct the pore structure of the heterogeneous catalyst [35].
Lines 245-249: In addition to sulfuric acid, researchers also tried to use nitric acid, hydrochloric acid, and phosphoric acid for the catalytic production of furfural. As strong mineral acids, nitric acid could promote the hydrolysis reaction. However, its promotion of the dehydration reaction was not enough, which could be due to the interaction between monosaccharides to form various by-products [60].
Revision parts:
Lines 208-210: “Previous studies have shown that the disappearance rate of furfural follows the first-order decomposition in the aqueous phase.” was revised to “Assuming that furfural does not participate in side reactions, the kinetic model for furfural formation aligns with that of hemicellulose hydrolysis, conforming to a pseudo-first-order model in homogeneous systems and a second-order model in heterogeneous systems [26, 36].”
Issue 6:
Is reference 39 also applicable to Lines 271-280? Otherwise provide reference(s) for those Lines.
Response
Thanks for your comment.
Reference 39 is applicable to lines 271-280, and the descriptions provided in these lines are sourced from Reference 39.
Issue 7:
Also add reference(s) for lines 404-407.
Response
Thanks for your comment.
We have supplemented the references.
Supplementary parts:
Due to the limited mass transfer, the actual reaction efficiency of metal oxides in the furfural production is lower than that of homogeneous catalysts, which is reflected in the higher catalyst loading and the lower furfural yield [101].
Issue 8:
The authors should really elaborate on the recyclability of solid catalysts for the production of furfural from biomass. For several of the mentioned solid catalysts there is stated that they have “strong” recyclability. However, working with solid biomass and/or the formation of humins this can be not that straightforward. Please elaborate in the text.
Response
Thanks for your comment.
The recyclability of heterogeneous catalysts depends on the absence of residual solid biomass or humins formation after the reaction. This study prioritizes high-performance solid catalysts that produce no organic solid residues, ensuring excellent recyclability. We have supplemented this premise for clarity. Additionally, Section 8 proposes targeted solutions for challenges in separating heterogeneous catalysts from solid residues or humins, including magnetic separation, high-temperature calcination, and solvent washing.
Revision parts:
Line 359: “Heterogeneous catalysts and reactants exist in different phases, thus endowing heterogeneous catalysts with significant recoverability.” was revised to “Heterogeneous catalysts have excellent recyclability, which depends on the absence of residual solid biomass or humins formation after the reaction.”
Issue 9:
Line 538: space missing.
Response
Thanks for your comment.
We have added spaces.
Supplementary parts:
Under the optimum reaction conditions (170 °C and 2 h), the conversion rate of xylose reached 97.5%, while the yield of furfural was 92.2%. The excellent catalytic performance of the catalyst can be attributed to its three-dimensional structure, which facilitates the diffusion of xylose and furfural.
Issue 10:
Line 591 and 594: what is meant by “workshop”?
Response
Thanks for your comment.
“Workshop” refers to a unit that completes certain production processes.
Issue 11:
Please provide a reference for Lines 649-650, Lines 704-707, Lines 725-726, Lines 834-847.
Response
Thanks for your comment.
We have supplemented the references.
Supplementary parts:
Lines 649-650: FA is considered to be the most important derivative of furfural, accounting for approximately 65% of its consumption [17q].
Lines 704-707: MOFs are considered as ideal supporters with molecular size screening function, and have great potential in FA production [175].
Lines 725-726: Some researchers believe that hydrogenation first occurs at the C=O position of furfural, which generates FA [131].
Lines 834-847: As shown in Figure 8, there are two primary pathways for the conversion of FA to MF. The first pathway is the direct route (Figure 8a). The metal sites serve as adsorption sites for hydrogen protons, which then initiate an attack on the C1 site, leading to cleavage of the C-OH bond and formation of MF [149].
Issue 12:
Line 893: remove “.” In gamma-valerolactone.
Response
Thanks for your comment.
We have removed “.”
Revision parts:
“γ-.Valerolactone” was revised to “γ-Valerolactone”
Issue 13:
Line 900-901: add reference(s).
Response
Thanks for your comment.
We have supplemented the references.
Supplementary parts:
Currently, the mechanism of upgrading furfural to GVL is still unclear, and the widely accepted mechanism is shown in Figure 9 [183].
Issue 14:
The authors mention a couple of times the usage of biphasic systems. Maybe this deserves a separate paragraph and a more in-depth discussion?
Response
Thanks for your comment.
The biphasic solvent system has demonstrated a significant positive effect on improving furfural yield, making it worthy of further exploration. Accordingly, we have added relevant content in the future prospects section.
Revision parts:
“From this point of view, the two-phase solvent system seems to be more conducive to the furfural production and upgrading. The target product produced from the reaction medium can enter the purified medium and prevent secondary degradation. However, challenges remain in terms of separating target products and recovering solvents.” was revised to “Biphasic solvent systems offer enhanced selectivity for furfural extraction by introducing an organic phase that is incompatible with water, effectively "capturing" furfural. This setup allows furfural to swiftly transfer from the aqueous to the organic phase, minimizing side reactions and thereby increasing yield and purity. Despite these advantages, biphasic systems face several key challenges. Slow mass transfer rates due to interfacial tension and phase transfer limitations can reduce overall reaction efficiency. Additionally, the separation and recovery of furfural from the organic phase remain complex, necessitating more efficient, cost-effective techniques. The use of organic solvents also poses potential environmental risks if not properly managed. Green solvents, such as ionic liquids, supercritical fluids, and DESs, offer promising, sustainable alternatives with advantages in efficiency, renewability, and lower environmental impact for furfural separation and purification.”
Issue 15:
Section 8.2.2: please also address the recyclability issue of solid catalysts.
Response
Thanks for your comment.
The recyclability of solid catalysts is addressed in future prospects section.
Supplementary parts:
8.2. Future Prospects:
- Enhancing the Recoverability and Reusability of Solid Acid Catalysts
Solid acid catalysts possess a notable advantage: they can be easily separated from soluble reaction systems through simple physical methods. This process is not only efficient but also significantly reduces environmental pollution and the burden of waste management. However, insufficient catalytic activity can lead to decreased selectivity for target products and the generation of by-products. These by-products can obstruct mass transfer channels and mask active sites, diminishing the catalyst's performance during recovery and reuse. Moreover, when solid acid catalysts encounter complex substrates, particularly biomass feedstocks, their incomplete degradation poses additional challenges, complicating catalyst recovery. To enhance the stability and recyclability of solid acid catalysts in complex catalytic systems, we propose several strategies. First, optimizing catalyst design by adjusting the distribution of active sites and improving structural stability can significantly improve catalytic efficiency and durability for furfural production. Second, developing efficient recovery technologies, such as magnetic modification for rapid separation, will facilitate effective catalyst recovery. Third, implementing secondary treatment methods, including calcination in an inert atmosphere or immersion in organic solvents, can effectively remove contaminants from the catalyst surface, enabling regeneration and recycling. Furthermore, incorporating smart materials, like self-cleaning surfaces or responsive polymer coatings, enables dynamic control of catalytic activity and streamlines subsequent recovery processes. These combined approaches are expected to promote the sustainable utilization of solid acid catalysts.